behaviour/ecology

bats, *Desmodus*, *Molossus*, *Eptesicus*, roosting ecology, sensory ecology

**Author for correspondence:**
Bridget K. G. Brown
e-mail: brown.6531@buckeyemail.osu.edu

# Do bats use guano and urine stains to find new roosts? Tests with three group-living bats

Bridget K. G. Brown[1], Lauren Leffer[3], Yesenia Valverde[2], Nia Toshkova[2,4], Jessica Nystrom[1], Rachel A. Page[2] and Gerald G. Carter[1,2]

[1]Department of Evolution, Ecology, and Organismal Biology, The Ohio State University, 318 W. 12th Avenue, Columbus, OH, USA
[2]Smithsonian Tropical Research Institute, Apartado 0843-03092, Balboa, Ancón, Republic of Panama
[3]University of Maryland, 4094 Campus Drive, College Park, MD, USA
[4]National Museum of Natural History at the Bulgarian Academy of Science, 1000 Sofia Center, Sofia, Bulgaria

 BKGB, 0000-0001-9774-3887; RAP, 0000-0001-7072-0669; GGC, 0000-0001-6933-5501

Many animals use social cues to find refuges. Bats can find roosts using the echolocation and social calls of conspecifics, but they might also use scent cues, a possibility which is less studied. The entrances of bat roosts are often marked by guano and urine, providing possible scent cues. We conducted eight experiments to test whether bats use the scent of guano and urine to find potential roosts. In field experiments, we tested if *Molossus molossus* (velvety free-tailed bats) in Panama and *Eptesicus fuscus* (big brown bats) in Ohio would investigate artificial roost boxes that were scented with guano and urine more often than a paired unscented control. We did not detect any difference in flights near the scented versus unscented roosts, and we detected only one entrance into any artificial roost (scented). In six captive experiments, we tested for the attraction of *Desmodus rotundus* (common vampire bats) and *Molossus molossus* to areas scented with guano and urine, under several conditions. Results were mixed, but overall suggested that the scent of guano and urine does not act as a strong lure for the tested bat species. We suggest that further tests of olfaction-based roost choice in bats should manipulate existing scent cues on familiar roosts.

## 1. Introduction

A key benefit of living in groups is that individuals can gain social information about resources, such as food or roosts [1]. For bats, potential roosts are often limited, and social information from conspecifics can help bats find new roosts [2]. In some species

and at some times of the year, roost-searching behaviour occurs frequently. For example, disc-winged bats roost in young furled leaves and move to new roosts daily [3]. To find unfamiliar roosts, bats can rely on the echolocation and social calls produced by conspecifics [3–7]. However, it remains unclear to what extent bats find and select roosts using other social cues, such as chemical cues.

There are several reasons why we might expect bats to use scent to locate roosts. Bats use scent cues for mate selection [8], foraging [9] and social recognition [10–13]. Compared with high-frequency bat calls, chemical cues can potentially travel farther distances, especially in cluttered forests, and they remain over time [14]. Predators and dogs can use guano and urine cues to locate bat roosts [15–17]. In at least one bat species (*Hipposideros speoris*), males appeared to use urine to relocate roosting spots and establish territory boundaries within a cave [18]. Some anecdotal reports suggest that guano stains can attract bats to a bat house, though others have argued that such stains have no effect [19,20]. If bats can be attracted to artificial roosts using a chemical lure, this could have important implications for bat conservation and management.

In this study, we focused on three common group-living bat species, *Molossus molossus* (velvety free-tailed bats), *Eptesicus fuscus* (big brown bats) and *Desmodus rotundus* (common vampire bats) for several reasons. First, they form stable social networks with preferred associations [21–24]. *Eptesicus fuscus* and *D. rotundus* also frequently switch roosts or roosting locations [24,25]. Second, these species live within communal roosts that are often stained with guano and urine. *Desmodus rotundus*, in particular, often live in roosts that are uniquely odorous because their diet of blood leads to copious urine and tar-like guano such that their pungent roosts can be detected at a distance by dogs or humans [15,16]. All three species will also roost in anthropogenic structures, showing some plasticity in their roost choices.

Third, our study species are likely to use olfaction in social interactions. *Desmodus rotundus* is attracted to the scent of prey [26], appears to use scent to aid in social recognition [27] and possesses an intact vomeronasal system and roughly twice as many intact vomeronasal type-1 receptor genes as other bats [28]. *Molossus molossus* and *E. fuscus* also have a vomeronasal organ indicating a capacity to detect social signals in pheromones, but they lack a vomeronasal epithelial tube and an accessory olfactory bulb, suggesting a less developed olfactory system than other mammals [29]. Despite this, *E. fuscus* can discriminate between familiar and non-familiar conspecifics using scent [10]. Though olfaction is not well-studied in *M. molossus*, the related Brazilian free-tailed bat, *Tadarida brasiliensis*, uses conspecific scent to locate and identify its pups, roostmates and roosting sites [30,31].

We conducted eight experiments to test whether bats are more likely to visit roosting sites when they are stained with guano and urine: a field experiment with *M. molossus*, a field experiment with *E. fuscus*, five captive experiments with *D. rotundus* and a captive experiment with *M. molossus*. In two of these experiments, we also presented playbacks of bat calls to compare the attraction of bats to chemical cues versus acoustic cues, and to test if the multimodal combination of cues would be a stronger lure than a unimodal cue. To draw an overall conclusion about the role of scent in roost-finding across all captive experiments, we estimated the size and precision of the mean bias towards scented roosts. We predicted that if bats use scent to locate roosts, then they would approach and spend more time investigating locations stained with guano and urine, compared with unscented locations. We also predicted that bats would be more attracted to acoustic cues than to scent cues, and that a combination of both cues might be even more attractive than either cue alone.

# 2. Methods for field experiments

## 2.1. *Molossus molossus* field experiment

To count visits to possible roosts and measure nearby bat activity for *M. molossus*, we created four artificial roost boxes ($30 \times 18 \times 10$ cm, pine wood, stained dark brown and sealed with clear silicone), each containing an ultrasonic microphone of a bat detector mounted in the roof interior and pointed downwards towards the entrance. Echolocation calls inside and nearby each box were recorded using a Wildlife Acoustic bat detector (SMMU2 and SM4+, Wildlife Acoustics, Maynard, MA) positioned inside the roost box with a 0 dB gain, a 16 kHz high-pass filter, a 256 kHz sampling rate, a 1.5 ms minimum wav file duration, a minimum trigger frequency of 20 kHz at 12 dB, a 3 s trigger window and a maximum call length of 15 s.

We deployed the four boxes as two pairs, one scented and one unscented, for one week at two different sites (13 weeks, 16 sites) from 10 May to 11 August 2019 in Gamboa, Panama. Most sites were near a known existing roost of *M. molossus*, within a distance of about 6 m (11 sites), 30 m (four

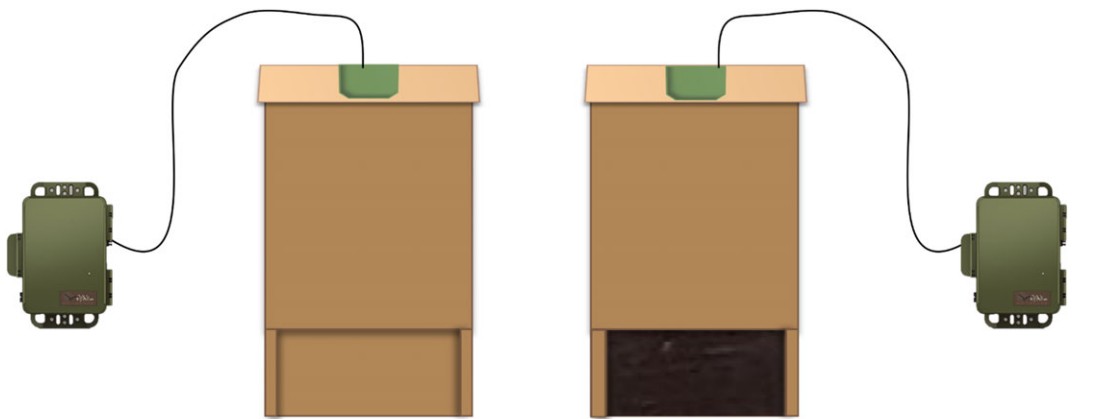

**Figure 1.** Field experiment set-up for testing bats' preference for scented and unscented roosts. Two roost boxes were placed 0.75 m apart and approximately 4 m from the ground at 16 sites in Panama and seven sites in Ohio. The roost box on the left is unscented, and the one on the right is scented with guano applied to a removable landing pad. Microphones in the roofs of the bat boxes carry acoustics signals to two SM4 + bat detector units that digitize and store the data.

sites) or 110 m (one site). After eight weeks, when all sites were used once, we used nine of the sites again. At each site, the paired boxes were mounted 0.75 m apart and approximately 4 m from the ground (figure 1). Each week, we swapped the scent cue treatment between the boxes. To create scent cues, we mixed 1 g of guano with 10 g of water and painted this on a cardboard landing pad attached below the opening of the scented roost box. The unscented box had a clean cardboard landing pad. We collected guano from four occupied roosts of *M. molossus*. For each pair of artificial roosts, we selected guano from the occupied roost that was located closest to the experiment site. The age of the guano collected from the roost was unknown, but it represents a natural sample of what was present at the roost entrance. Guano samples were stored in sealed bags at room temperature and refrigerated for extended periods to prevent further change in microbial composition within the bag. The landing pads were removed and replaced at each deployment (each week) and the scent cue was re-applied.

To test the effect of acoustic cues on bat activity near the boxes, we conducted playbacks on one random night per week, except for the first or last night of a week. On the playback night, we mounted an ultrasonic speaker (Avisoft USG Player BL Pro, Avisoft Bioacoustics, Berlin, Germany) between the two experimental roosts and broadcast echolocation and social calls for 1 h immediately after sunset, when these bats are active outside the roost [32]. We constructed playbacks from recordings of calls from flying bats ($n = 22$) that were released from a platform and calling towards the microphone (Avisoft ultrasonic condenser microphone, digitized with an UltraSoundGate 116, 250 kHz sampling rate, 16-bit resolution) from a previous study conducted at the same location. In this previous study, the bats were captured in mist-nets and were fitted with pit-tags before being released [33]. The playback was a 427 s recording with 10 in-flight social calls and 122 echolocation calls that we played on loop for an hour. To preserve the original temporal call pattern, we kept sequences intact when constructing playbacks in Avisoft SASLab Pro. We spaced recorded samples with 8 s of silence to simulate natural breaks in calling rates. To remove sounds below 20 kHz and above 100 kHz, we used a high-pass filter and a time-domain filter. We also removed calls that were either less than 15% or reached 100% of the maximum possible amplitude.

To count echolocation calls of *M. molossus*, we used Avisoft SASLab Pro to detect possible bat calls using a single amplitude threshold of −70 dB and to measure the onset time, duration, frequency at the peak amplitude and peak-to-peak amplitude. To select *M. molossus* calls, we then used R to select cases that were 30–50 kHz, 1–8 ms in duration and occurred during the activity period of *M. molossus*, between 18.00 and 21.00 h. Recording sites were near known roosts of *M. molossus*, which is also the most common *Molossus* species in this area.

## 2.2. *Eptesicus fuscus* field experiment

We conducted a similar field experiment targeting *E. fuscus* in Ohio. As in the Panama field experiment, we deployed two pairs of artificial roost boxes with bat detectors for one week at two sites. Boxes were stained black to allow for more heat retention but were otherwise the same as described above (figure 1).

For the scent cue, we painted a mixture of 5 g of guano and 10 g of water on to the removable cardboard landing pads. We used more guano than in the previous test due to higher availability of guano. We collected guano from three different big brown bat colonies and alternated the guano source each experimental week ($n = 17$ weeks). The control box was left unscented. The two pairs of boxes were rotated between seven sites with known big brown bat activity in Columbus Metro Parks near Columbus, Ohio from 15 April to 12 August 2019. Each site was used six times.

To count echolocation calls and possible visits of *E. fuscus*, we used the same procedures described above for the *M. molossus* field experiment. To target *E. fuscus* calls, we selected calls across the whole night that had peak frequencies of 25–40 kHz and durations of 5–20 ms [34]. We did not conduct any playback experiments in Ohio.

## 2.3. Statistical analysis for field experiments

First, to check for clear roost 'visits' by bats, we looked for clipped calls (i.e. those that exceeded the maximum possible amplitude) in each of the paired boxes. We knew that clipped calls would indicate that a bat was on the landing pad and calling directly into the microphone inside that box based on recording three captive *E. fuscus* (from the Ohio Wildlife Center) that were placed onto the landing pad of the roost box.

Second, to investigate whether the highest echolocation intensity calls (i.e. closest approaches) were more likely to be recorded at the scented box than the unscented box, we manually inspected the spectrograms of bat calls with amplitude over 5% of the maximum possible amplitude. To ensure calls were independent bat approaches, we only counted the loudest calls separated by 10 min of each other. This is a fairly conservative approach, but our conclusions do not change when instead we used a separation time of 1 min or when not excluding any calls. We then conducted a binomial test to test whether the loudest bat calls were more likely to be recorded at the scented box.

Third, to assess bat activity near each box, we counted the number of possible bat echolocation calls at the scented box and the unscented box within each night and site. We then measured the scent bias as the difference between the number of calls at each box (scented – unscented) divided by the total number of calls at both boxes (scented + unscented) for each independent site. To calculate a non-parametric 95% confidence interval (CI) around the bias for each experiment, we bootstrapped the mean scent biases using the 'boot' package [35] in R (5000 iterations, 'boot' package, percentile method).

# 3. Results for field experiments

In Ohio, we did not detect any clipped calls (100% maximum amplitude), but we detected one clear case of an approach: an extreme outlier in call intensity (i.e. 99.75% of the maximum amplitude compared with the next loudest call, which was 58% of the amplitude and 508 896 other values which ranged from 3.7 to 27% of the maximum amplitude). This case suggested that only one *E. fuscus* echolocated directly into a scented roost box. We did not detect a difference in the total number of calls at the scented and unscented boxes (two sites, five weeks, permuted paired *t*-test: $t = 0.58$, $p = 0.5$). Only 11 of the 27 closest bat passes (i.e. loudest call bouts) were at the scented box (binomial test: $p = 0.44$).

In Panama, we did not detect any evidence for visits from *M. molossus* at either box. We did not detect more calls at the scented boxes; in fact, the sample mean was trending to be higher at the unscented control box (six sites, six weeks, permuted paired *t*-test: $t = -1.86$, $p = 0.08$). Only eight of the 24 closest bat passes (loudest call bouts) were at the scented box (binomial test: $p = 0.15$). We also did not detect more calls during the playback hours (432 calls per hour, 95% CI: [274, 583]) than during the corresponding non-playback hours (505 calls per hour [311, 677]).

# 4. Methods for captive experiments

## 4.1. Captive Experiment 1: *Desmodus rotundus* choosing dark locations to hide from light

We conducted five experiments with captive vampire bats, *D. rotundus*, each with different observers, and each improving on the methods of the previous test. In the first experiment, we repeatedly tested five captive adults (four females and one male, originally from Chicago Brookfield Zoo), by presenting them with a scented and an unscented refuge, which consisted of rolls of plastic mesh covered in paper and black plastic (figure 2*a*). The scented tube contained a mesh bag at the top with

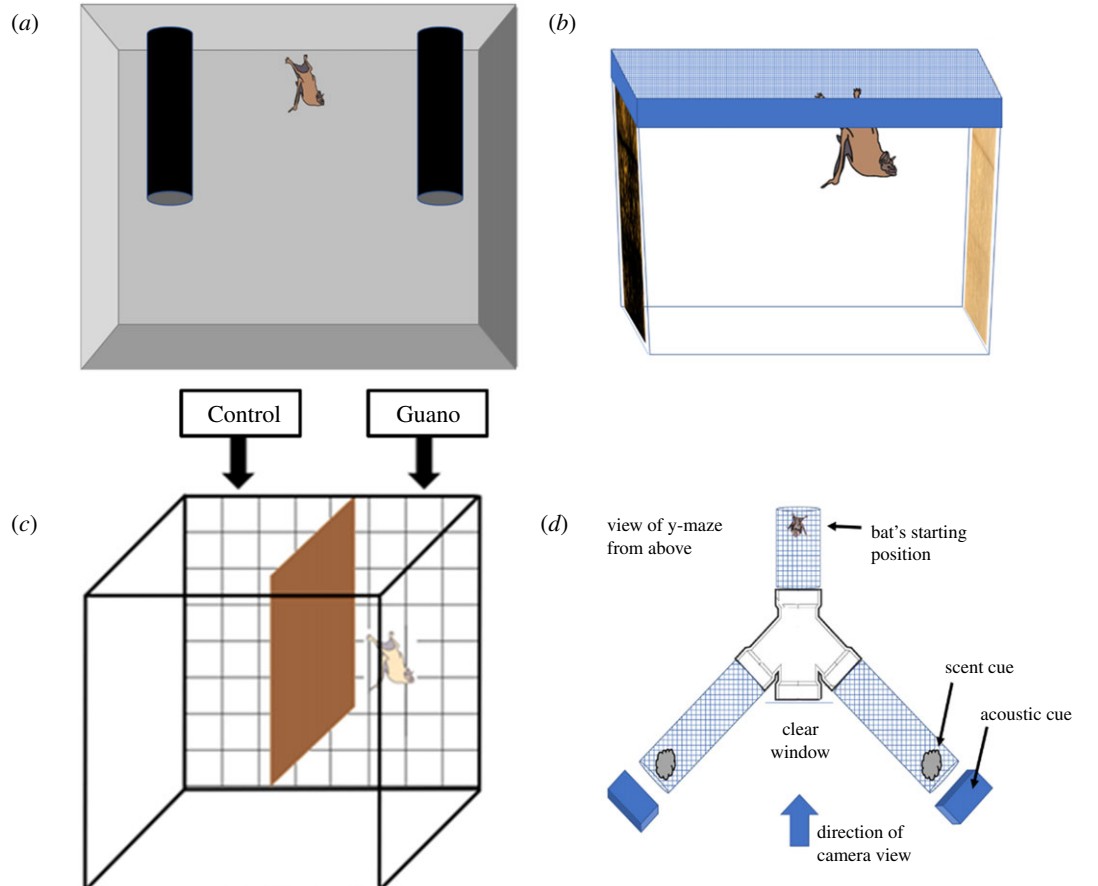

**Figure 2.** Captive experiment set-ups for testing bats' preference for scented and unscented roosts. Panels show the test chambers for (*a*) Captive Experiment 1 (*D. rotundus* choosing dark locations to hide from light), (*b*) Captive Experiments 2 and 3 (bats choosing scented or unscented surfaces to hang on), (*c*) Captive Experiment 4 and 6 (bats choosing scented or unscented surfaces under paper towel) and (*d*) Captive Experiment 5 (*D. rotundus* choosing to move towards acoustic cues, scent cues or their combination in a y-maze).

a paper towel and bedding soiled with guano and urine from the bat's home enclosure. The unscented tube contained unsoiled paper towels and bedding and was otherwise the same. To induce bats into choosing to roost inside one of the tubes, we used a dim light to ensure that the tubes were the darkest location to roost. We tested each bat individually and repeatedly in three small cages (0.3 × 0.3 × 0.4 m) and one flight cage (1.7 × 2.1 × 2.3 m). In the flight cage, we also conducted tests with one scented and three unscented tubes. Bats did not immediately enter the roost tubes and would often hide in a corner of the cage before moving into the darker roost tube. We therefore repeatedly checked their location at the next convenient time until the bat was inside a tube (1–24 h). Due to this haphazard sampling, we could not measure latency or be sure whether this choice was their first choice, but we suspect that, once a bat selected and hid inside a dark tube, it probably did not leave that refuge in the presence of ambient light. If the bat did not choose either tube, the cage was checked at least 1 h later. After all five bats made a choice, we swapped the bats into new test cages and swapped the locations of the scented tubes within each cage. Each bat made two to six roost choices in different test cages (*n* = 19 choices). The individual bat was the observational unit.

## 4.2. Captive Experiment 2: *Desmodus rotundus* choosing surfaces from which to hang

In Captive Experiment 2, we tested a greater number of bats (*n* = 20) than in Captive Experiment 1 (*n* = 5). We also used a test chamber that was familiar to the bats to remove possible neophobia that may have affected Captive Experiment 1. The bats tested were from a long-term captive colony at the Organization for Bat Conservation in Bloomfield Hills, Michigan (described in [27]). To test for a preference to hang near or on a scented surface, we attached a clean paper towel sheet to one side of the tank and a

paper towel sheet stained with urine and guano on the other side. The guano and urine were collected daily from the floor below the captive colony. The bat could hang from the scented side, the unscented side or the ceiling, but it could not hang from the remaining two clear sides of the test chamber that were smooth (figure 2b). We left an individual bat in the chamber in the dark, and then opportunistically sampled the position of the bat after a period of 1–5 h. If the bat was on the floor or ceiling of the chamber, we took another sample at least 1 h later. We swapped the location of the scented towel and cleaned the chamber between each tested bat. Each bat made one to nine choices ($n = 48$ choices). The individual bat was the observational unit.

## 4.3. Captive Experiment 3: *Desmodus rotundus* choosing surfaces from which to hang

For Captive Experiment 3, we replicated Captive Experiment 2 using 22 *D. rotundus* (14 wild-caught adult females, four captive-born males, four captive-born females) in Panama, but here we recorded each individual bat using infrared light and a Sony Nightshot camcorder (hereon 'infrared video'). We then sampled the roosting position of the bat every 10 min for 4 h within the chamber (from midnight to 04.00 h). During this time, bats in the colony are likely to be active and at rest. Choices were defined as a bat touching the scented or unscented side. Each bat made one to three choices ($n = 39$ choices). The individual bat was the observational unit.

## 4.4. Captive Experiment 4: *Desmodus rotundus* choosing between spaces in divided chamber

In Captive Experiment 4, we attempted to remove possible visual and echoacoustic cues caused by guano on the paper towel (e.g. a darker surface with more texture) by covering the scented paper towel with two clean paper towels and placed three clean paper towels in a similar manner on the other side. We tested 33 different adult *D. rotundus* (28 females and five males) captured in Panama. Bats were placed in a test chamber ($40 \times 56 \times 40$ cm, glass) and could choose to hang on one of two sides divided by a cardboard divider covered in plastic (figure 2c). The scent cues were 5 g of guano and urine collected every one to three days from a captive vampire bat colony. Between each trial, we removed the paper towels, cleaned the entire chamber and swapped the location of the scent treatment. To avoid potential biases caused by airflow or lighting, we also rotated the position of the test chamber inside the room between trials. We recorded the trials using infrared video. Bats were held in a cloth bag, which we placed in the centre of the chamber below the divider facing towards the back wall, allowing the bat to crawl out of the bag on their own. We tested each bat one to four times in 91 trials (48 trials that were 1 h and 43 trials that were 4 h). The individual bat was the observational unit.

## 4.5. Captive Experiment 5: *Desmodus rotundus* choosing to move towards acoustic cues, scent cues or their combination in a y-maze

In Captive Experiment 5, we compared the immediate attraction of bats to either roost sounds, roost scents or their combination, using a y-maze with interchangeable arms (figure 2d). We used 45 *D. rotundus* (26 adult females, 16 adult males, two juvenile females and one juvenile male) from the same colony we used for Captive Experiment 3. The bat could walk down the central junction toward either the arm with the stimulus or the arm with the control. The left and right arms led to plastic-grid-mesh tubular arms wrapped in clear plastic wrap with the ends left open. The central opening of the junction was covered with clear plastic to allow the bat's movements to be recorded by video. Bats were introduced into the back opening. The experimenter and camera were centred to prevent side bias.

We conducted three trial types. In a scent trial, a bat chose between maze arms containing either a balled-up paper towel scented with guano and urine from the colony (test arm) or a balled-up unscented paper towel (control arm) at the end. In a sound trial, a bat chose between maze arms leading either towards a playback speaker playing a recording from a wild *D. rotundus* roost (test arm, details below) or a cardboard box of the same size as the speaker (control arm). In a combination trial, we presented both the acoustic and scent cues in the test arm and both control cues in the control arm. At the start of each trial, we placed the bat inside the y-maze and filmed for 300 s with infrared video to record its movement. If the bat did not enter either arm in this time, a new test with the same bat was started immediately. Between bats, the arms of the maze were swapped. If a bat urinated or defecated in an arm, both arms were cleaned, and the plastic wrap was replaced before testing the next bat. All 45

bats were assigned to a trial type in a random, balanced design, and 35 of the bats were also tested in the two other trial types in random order. We therefore analysed both first choices ($n = 15$ sound, 15 scent and 15 combination trials) and all choices ($n = 39$ scent, 38 sound and 38 combination trials). As responses, we measured the difference in (i) duration of time that the bat spent in each arm and (ii) decision speed, which is the trial duration minus the latency until the bat entered an arm.

To create the acoustic cues used, the echolocation and social calls emitted from a *D. rotundus* roost were recorded. We placed an Avisoft CM16 ultrasound condenser microphone (frequency range 10–200 kHz, connected to an Avisoft UltraSoundGate 116, constant gain, 16-bit resolution, 250 kHz sampling rate) in the entrance to a large hollow tree in Tolé, Panama that contained more than 200 *D. rotundus*, and from which 29 of the 45 test subjects were captured. To create playbacks, we compiled five 60 s samples of time when no sounds reached maximum amplitude. During each trial, we played the five acoustic samples in random order from an Avisoft USG Player BL Pro ultrasonic speaker. To create scent cues, we used paper towels to gather guano and urine each day from below the captive colony.

## 4.6. Captive Experiment 6: *Molossus molossus* choosing between spaces in divided chamber

In Captive Experiment 6, we tested an attraction to guano and urine in captive *M. molossus* (10 males and eight females) from three different roosts in Gamboa, Panama. To create scent cues, we wetted 2–3 g of guano with 5 g of water, collected from the roost where the subject bat was captured and spread on the paper towels. All trials were 4 h and bats were each tested once. The procedure was otherwise identical to Captive Experiment 4. After being given water and mealworms, bats were released the same night they were captured.

## 4.7. Statistical analysis for captive experiments

To calculate an effect size, we used a standardized statistic that could be applied to the data from all captive experiments. For each subject within each experiment, we first calculated the proportion of a bat's total choices (or time) on the scented side, $P_o$, and the proportion expected by chance, $P_e$. We then defined the 'scent bias' as $(P_o - P_e)/(1 - P_e)$. The scent bias therefore ranges from values of $-1$ (all sampled time on the unscented side) to 1 (all sampled time on the scented side). We estimated the effect size as the mean scent bias across all bats for each experiment. To calculate a non-parametric 95% confidence interval (CI) around the bias for each experiment, we bootstrapped the mean scent biases using the 'boot' package [35]) in R (5000 iterations, 'boot' package, percentile method).

To get an overall bias across the captive experiments, we used two different methods. First, we calculated a weighted mean of the experiment biases scaled by the number of bats used in each experiment and bootstrapped these six weighted means to get a 95% CI around the grand mean. We called this the 'conservative' estimate. Second, we calculated the overall mean bias by bootstrapping all the observed scent biases for each bat across all captive experiments. This approach assumes that choices by bats in different experiments were independent observations. We call this estimate 'non-conservative'.

## 4.8. Comparing scent, sound and bimodal treatments in Captive Experiment 5

For the choices of the bats in the y-maze trials of Captive Experiment 5, we used the bootstrapping procedure describe above to calculate 95% CIs for the treatment bias for scent, sound and their combination. To calculate *p*-values, we used permutation tests where we compared the observed mean difference in speed or duration to 5000 expected differences when the response data were randomly swapped between arms within each trial and between trial types within each bat (i.e. data were never permuted between bats). Finally, we used paired permuted *t*-tests to test if the within-bat response to trials with sound was greater than trials without sound, and if the within-bat response to the scent and sound combination trial was greater than the sound trial alone.

# 5. Results for captive experiments

Overall, we did not detect a consistent and strong bias towards roosting in locations scented with guano and urine for the five experiments with *D. rotundus* (weighted mean = 0.18, 95% CI: [−0.06, 0.38]) or for

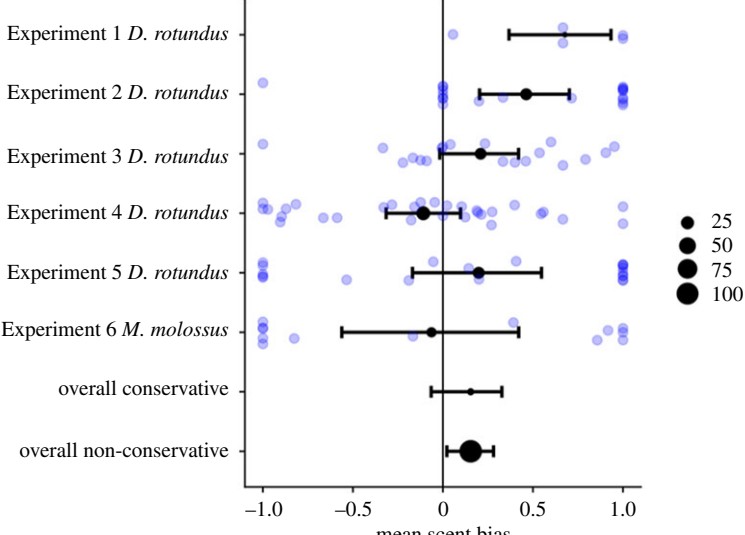

**Figure 3.** Scent bias estimates in captive experiments. The distribution of scent biases and the means and 95% CIs are shown for Captive Experiments 1 to 6. The size of the circles in the figure is representative of the number of bats used for that experiment. The overall conservative estimate was based on the weighted means from each test ($n = 6$). The overall non-conservative estimate was based on the standardized biases shown by each bat across all tests ($n = 107$).

all six captive experiments (conservative weighted mean bias = 0.15, 95% CI: [−0.06, 0.32]; non-conservative weighted mean bias = 0.15, 95% CI: [0.02, 0.29]).

Results from all captive experiments are shown in figure 3. In Captive Experiment 1 (*D. rotundus* choosing dark locations to hide from light), the five bats often chose not to enter either a scented or an unscented roost tube, but when they did choose to enter a roost tube, all five bats chose scented tubes more often (figure 3). Bats chose to roost in the scented tube over the unscented tube in nine of 10 choices in the small cages with two tubes, and in five of eight choices in the larger flight cage trials with one scented roost and three unscented roosts. If we assume these choices were unbiased and independent, the binomial probability of seeing these outcomes by chance is only 0.02 and 0.03, respectively.

In Captive Experiment 2, bats were more likely to hang on or at the surface stained with guano and urine than on the unscented surface; however, when we attempted to replicate this test with a group of wild-caught bats (Captive Experiment 3), we no longer detected such a clear bias. In Captive Experiment 4, we removed possible visual and echoacoustic cues and used an altered set-up, and we detected no evidence for a scent bias. When we repeated Captive Experiment 4 with *M. molossus* instead of *D. rotundus* (Captive Experiment 6), we did not see a bias toward either the scented or unscented side.

Finally, in Captive Experiment 5, we compared with immediate attraction of wild-caught *D. rotundus* from Panama to scent or sound cues. Isolated *D. rotundus* in a y-maze spent 77 s longer (permutation test: $n = 38$, $p < 0.002$) and moved 108 s faster ($p < 0.002$) into the arm containing the sounds of a roost, compared with an empty arm. By contrast, the bats did not show a clear bias towards the scented arm ($n = 39$, duration bias = 11 s, $p = 0.33$; speed bias = 27 s, $p = 0.19$). These results did not differ when considering only the first choice by each bat. The attraction of the bats to the simultaneous sound and scent cues ($n = 38$, duration bias = 121 [78, 199] s, $p = 0.33$; speed difference = 141 [91, 194] s, $p = 0.19$) was not greater than the attraction towards the sound cue alone (figure 4; paired permuted *t*-test, duration: $t = 1.78$, d.f. = 34, $p = 0.08$; speed: $t = 1.54$, d.f. = 34, $p = 0.14$).

# 6. Discussion

## 6.1. Field experiments

We could not compare the visitation rates to the treated versus untreated bat houses because the boxes logged only a single visit. The prior probability of inspections by bats was unclear. We deployed our scented and unscented roost boxes for one week at a time at seven sites with relatively high bat activity near Columbus, Ohio and at 16 buildings in Gamboa, Panama, but bats almost never

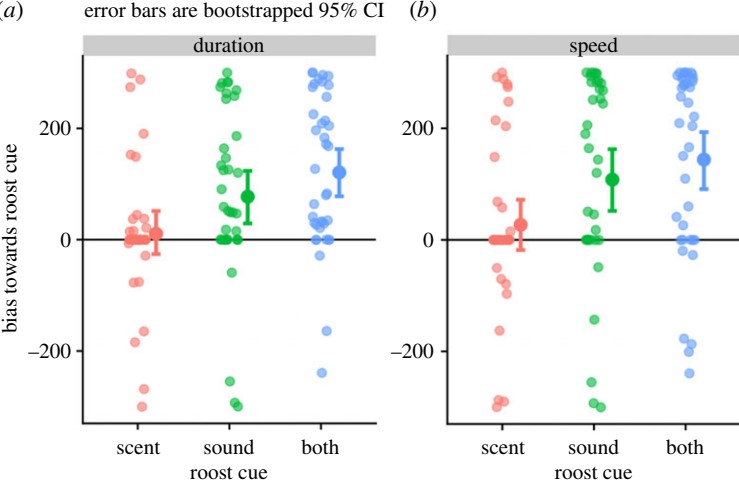

**Figure 4.** *Desmodus rotundus* are attracted to roost sounds, but not scents. Compared with an empty arm, isolated *D. rotundus* in a y-maze were attracted to the arm with a sound cue or a scent and sound cue, but not towards the scent alone cue. Error bars show the bootstrapped 95% CI around the mean bias. The within-bat mean bias was greater in trials with sound compared with the trials without sound (duration: $t = 3.73$, d.f. $= 34$, $p < 0.0002$; speed: $t = 3.53$, $p = 0.002$).

echolocated into the interior to check their suitability. This result is consistent with the notion that inspections by bats to new potential roosts are relatively rare or that they primarily occur at certain times of the year. For example, if we had deployed the experimental roosts in Ohio earlier, we may have had more inspections as bats first start to return to their summer foraging sites.

Despite the lack of visits, the field experiments do suggest that painting bat houses with guano does not make them extremely attractive. We did not find clear evidence that guano and urine can act as a strong chemical lure for artificial bat roosts. Occupancy of bat houses is determined in part by whether bats discover it, which requires that they at least inspect the roost. Hence, our data suggest that the baseline rates of inspection of artificial bat houses are quite low. The artificial roosts may have not been inspected due in part to neophobia. Conducting this experiment over a longer period of time with each roost being deployed at a particular location for a more than one week may have resulted in more roost visits. Also, if the artificial roosts smelled distinctive and novel to the bats, any aversion to this odour may have overshadowed any attraction to the guano and urine scents.

Another caveat is that the relatively weak scent cues at our roost boxes might have been masked by existing scent cues from nearby occupied roosts. This possible effect may also explain why we did not detect more activity during the hours when we conducted playbacks. The *M. molossus* at these sites quickly depart and return from their roosts using relatively fast and straight flight paths. With many bats calling nearby as they exit and enter the occupied roost, the playbacks of the echolocation calls may not have been salient. During a previous experiment, the activity of *M. molossus* in Panama increased in response to the highly salient playbacks of distress calls [36], but not to playbacks of echolocation or social calls (G. Carter 2015, personal observation).

Finally, the bats might detect the difference between familiar and unfamiliar conspecific guano, and unfamiliar social cues might either attract or deter them. The *D. rotundus* were presented with familiar scent cues. However, the *Molossus* field experiment used guano that was collected from nearby occupied roosts, and the *Eptesicus* field experiment used unfamiliar scent cues. If the unfamiliar scent of guano actually deters bats from inspecting a potential roost, we may have even unintentionally warded them away from our experimental roosts.

## 6.2. Captive experiments

The results of the captive experiments are consistent with three possible general interpretations. The first possibility is that isolated *D. rotundus* are not attracted at all to guano and scent cues, and the first two experiments yielded false positives. The observations sampled in Captive Experiments 1 (*D. rotundus* choosing dark locations to hide from light) and 2 (*D. rotundus* choosing surfaces from which to hang) were haphazard and not filmed. We attempted to improve on the methods of these earlier tests by filming the responses, testing more subjects and using more controlled procedures. For example, in

Captive Experiment 4 (*D. rotundus* choosing between spaces in divided chamber), when we removed possible visual or echoacoustic cues, we did not detect a scent bias. If bats preferred to roost on a more echoacoustically textured surface, then this could lead to the false appearance of a scent bias (at least in Captive Experiment 2). When conditions are not well-controlled, it can be easy to misidentify which sensory cues are used in decision-making.

A second different explanation is that isolated *D. rotundus* are attracted to guano and scent cues, but only under certain circumstances, such as when seeking a dark refuge or when choosing a place to sleep, but not when exploring a novel environment. For example, if the role of scent cues depends on the spatial scale and concentration gradient of the chemical cue, then bats might not use scent cues in exploring an area with a diffuse odour, such as in Captive Experiments 2 and 3 (bats choosing surfaces from which to hang), and 4 (bats choosing between spaces in the divided chamber). However, in the absence of other cues, they might choose to enter a scented hole among several in a tree, similar to Captive Experiment 1 (bats choosing dark locations to hide from light). Time of day might also play a role. For example, Captive Experiment 3 occurred during the second half of the night when bats would normally not be sleeping within a roost.

A third explanation is that isolated *D. rotundus* demonstrated a bias towards the familiar scent of guano and urine in the earlier experiments, but only due to a simple familiarity bias (reduced neophobia), rather than an attraction to the chemical cues. While both roost tubes were initially novel in Captive Experiment 1, the familiar scent of guano and urine may have reduced neophobia in the scented tube, causing them to seek refuge from the light there. This bias towards the more familiar scent could then be exaggerated by repeatedly testing the same bats in different cages.

Although there are multiple possible explanations, all these interpretations suggest that the scent of guano and urine does not act as a reliable lure for these two species, when compared with other cues. A captive bat might use scent as a cue in the absence of other social cues, but this does not tell us how important this cue is relative to other cues in a natural context. The *D. rotundus* in Captive Experiment 5 chose between a scented and unscented tube in the form of the two arms of the y-maze, but here, where the entire setting was novel and dark, there was no scent bias in their exploration.

The mixed results of our captive experiments also highlight the importance of replication and non-selective publishing [37]. Given a mix of some positive and null results, authors can be improperly incentivized to cherry-pick results that provide a narrative that is more straightforward, clear and simple. Despite the flaws of the first two captive experiments, we report them because their results actually weaken our confidence in the possible interpretation that scent plays no strong role in roost-finding. Our aim here was to present and quantify the full uncertainty in how bats might use scent to choose a roosting location and to suggest improvements in future studies that might better resolve if and when bats use scent cues to locate roosts.

We considered *D. rotundus* to be a strong candidate for using the scent of roostmate guano and urine to select roosts, because they are highly social, have exceptionally odorous roosts and have well-developed olfactory systems compared with other bats, and some authors have even suggested that *D. rotundus* follow the scent of urine trials [16]. However, our results show that the scent of guano and urine is not as salient or attractive to *D. rotundus* as acoustic cues. The attraction of isolated *D. rotundus* to the echolocation and social calls of a roost is not surprising given that isolated *D. rotundus* are attracted to social calls [38], and that several other bat species are attracted to the acoustic cues at conspecifics at roosts [3–7,39,40].

Previous studies show that three European bat species also used acoustic cues but not scent cues [5,6]. One reason that bats might be more immediately attracted to calls compared with scents is that calls indicate the presence of a conspecific in the present moment, whereas a scent mark or a guano or urine stain is only a cue of a bat being present at some time in the past. We also expect that the chemical *cues* in guano and urine are also less salient than chemical *signals* from scent marking [41–44]. If the chemical composition of guano and urine varies over time or by context, then cues from guano and urine could be more attractive at certain times of production (e.g. urine produced at the end of night) or at certain locations (e.g. the entrance of a roost as opposed to inside of the roost). There could also be variation in how the cues are perceived (e.g. scent compounds in urine might be salient only during a certain season).

If guano could act as a chemical lure for bats in general, this could be a useful tool for bat conservation and management, but we detected no evidence that the scent of guano and urine acts as a lure to attract bats into new artificial roosts. Given our lack of visits in the field experiments, much uncertainty remains, and the role of scent as an informative habitat cue in bats is surprisingly understudied. For example, chemical analyses have revealed that social information is contained in urinary proteins in rodents, but similar analyses have not been done for bats.

Based on our results, we suggest that the best approach for future studies of scent is to manipulate existing scent cues for bats attempting to relocate familiar roosts (e.g. [18]). For instance, a captive colony of bats could be forced to roost inside of the middle of three adjacent roost boxes, by blocking the entrances to the left and right box. After all the bats have learned the location of this roost entrance, an experimenter could block the known roost entrance, move the scent cue to either the left or right, and force individual bats to choose between those alternative roost entrances. This design would test the bats during their motivational state of seeking to return to the roost and eliminate the problem of neophobia leading to fewer choices. If bats did not show any scent bias under those conditions, this would be fairly conclusive evidence that bats do not use scent in roost selection.

Ethics. This work was approved by the University of Maryland Institutional Animal Care and Use Committee (Protocol R-10-63), the Smithsonian Tropical Research Institute Animal Care and Use Committee (nos. 2015-0915-2018-A9 and 2019-0501-2022), the Panamanian Ministry of the Environment (nos. SE/A-76-16 and SE/A-36-19) and the Columbus and Franklin County Metro Parks through an access permit.

Data accessibility. Data and R code are available on Figshare (https://doi.org/10.6084/m9.figshare.12760271).

Authors' contributions. B.K.G.B., G.G.C., L.L., Y.V., N.T. and J.N. collected the data. B.K.G.B. and G.G.C. analysed the data. B.K.G.B. and G.G.C. drafted the manuscript, and all authors critically revised the manuscript. G.G.C. and R.A.P. supervised the work.

Competing interests. We declare we have no competing interests.

Funding. G.G.C. was supported by student research grants from the Animal Behaviour Society and the American Society of Mammalogists, a Doctoral Dissertation Improvement Grant from the National Science Foundation (1311336). L.L. was supported in part by a grant to the University of Maryland from the Howard Hughes Medical Institute through the Science Education Program. B.K.G.B. was supported by student research grants from Sigma Xi Grants-in-Aid of Research and Ohio State's Critical Difference for Women Professional Development Grant.

Acknowledgements. We thank R. Mies, A. Felk and J. Fabian for supporting and accessing to captive bat colonies and E. Jacobs for help with data collection. We thank the Smithsonian Tropical Research Institute for logistical support in Panama, in particular, G. Cohen, M. Dixon, A. Heckley, C. Santiago and S. Stockmaier. We also would like to express our gratitude towards J. Stancourt and M. Little for their assistance with data collection on the Ohio Field Experiment. We would like to provide further acknowledgement to the following people for helping to procure guano and roosting locations for the field experiments: The Ohio Wildlife Center, Columbus and Franklin County Metro Parks Staff (particularly C. Morrow), L. Cooper at NEOMED, and J. Kohles in Panama. Finally, we would like to thank J. Kohles for also providing playback calls for field experiments with *M. molossus*.

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
