## [Reviewer comments · Royal Society Open Science]

Review History

RSOS-201055.R0 (Original submission)

Review form: Reviewer 1 (Michael Smotherman)

Is the manuscript scientifically sound in its present form?

Yes

Are the interpretations and conclusions justified by the results?

Yes

Is the language acceptable?

Yes

Do you have any ethical concerns with this paper?

No

Have you any concerns about statistical analyses in this paper?

No

Recommendation?

Accept with minor revision (please list in comments)

Comments to the Author(s)

This paper describes a series of field and lab experiments designed to detect the influence of olfactory cues on roost selection. This addresses some long-standing assumptions in the field of bat behavioral ecology that haven't been thoroughly tested. Despite the predominantly negative results, the experiments were logical, appropriate and systematically executed. The paper lacks any clear conclusions, other than demonstrating that there is not a clear or obvious scent bias when bats select temporary roosting sites. The principle benefit of the manuscript is that others will likely want to bypass these experiments and may find the negative results useful for building upon. It still seems probable that bats use olfactory cues for roost selection, mate attraction or territorial behaviors, but it might take more sophisticated, time-intensive and expensive efforts to quantify them.

Only main complaint is the way the experiments are introduced in the methods is very confusing and the presentation pattern isn't consistent going into the results. The 8 experiments outlined in methods are broken down into 8 headered sections varying by species, location, field or lab, and experimental conditions. There are double experiment 1s and 2s, which gets dizzying in the results, which only has two sections, field and lab, and bounces back and forth between different experimental numbers, species and locations. I would suggest consecutively numbering all the experiments 1-8, while still giving each an appropriate descriptive title in the header. Then in Results keep the two sections format but systematically go through and refer to experiment 1, 2, 3 etc. This would be easier for me to follow than trying to keep track of which experiment 1 (captive or field) is being discussed.

Minor Comments

Page 5, line 2: *Molossus molossus* field experiments would be a more appropriate title for this section, and then Page 7 line 2 *Eptesicus fuscus* field experiments. The species distinction seems more significant and relevant than where they were performed.

Page 7 line 43: Choosing "clipped" as the threshold is ok, but please provide details about what the threshold amplitude would be (i.e. >100 dB SPL). Presumably the recorders maxed out at 5 or 10 volts, and with the microphone sensitivity it should be straight forward to convert clipping to a loudness threshold. ON the other hand, Results-Field Experiments (page 13 lines 50+) indicates that only one *Eptesicus* call at 95% (i.e. below clipping) was detected, and none for *Molossus*. If NO calls caused clipping, then what is the point of defining this parameter at all?

Page 8 line 1: 10 minutes is a long time. Would this exclude counting bats travelling in pairs or groups? I'm guessing that if it mattered the paper would have said so, but would be worth knowing whether or not shortening this to 1 minute (or less) would have produced different results.

Page 8 line 39: typo, the otherwise same?

Page 8 line 51: Using the camcorder in the later experiments seems like a no-brainer, but here the "opportunistically scored after 1-24 hours" leaves room for criticism. Was there at least a regular schedule for checking the tubes over the first 24 hours, or was it assumed that once a bat chose a tube that they would continue to use just that tube?

Page 9 line 4: Typo "In captive experiment two experiment, we tested...". Drop the second experiment.

Page 9 line 34, line 53, and elsewhere. This "Individual bat as the observational unit" statement is not a complete sentence and is repeated at the end of several sections.

Page 9, line 48, of captive experiment 3. At what time during the night were the bats assayed? It says every 10 minutes for four hours, which is fine, but when during the night did these four hours occur? Is there a problem here that the wild-caught bats would normally not want to roost during the night time, since they would normally be out feeding?

Page 9, Captive Experiment 4: How does covering a scented towel with a clean unscented towel control for echoacoustic cues? Need clarification on what the potential cues were.

Figures 3 and 4 are very interesting.

Discussion: While I generally agree that publishing negative results is useful where appropriate, the discussion goes overboard with this defense. Captive experiments 1 and 2 are messy and flawed and might fairly be considered unpublishable pilot experiments. The larger issue may be that short-term experiments like these may not be the best way to capture the role of olfaction in roost selection. As I'm sure the authors recognize, olfaction is likely just one of several cues getting integrated with seasonality, time of day/night, sex and reproductive status, stress and social factors etc. Still, collectively the eight experiments make a cohesive story that may save time for future investigators. Experiment 5 was very well done and would love to see that expanded upon.

Review form: Reviewer 2

Is the manuscript scientifically sound in its present form?

Yes

Are the interpretations and conclusions justified by the results?

Yes

Is the language acceptable?

Yes

Do you have any ethical concerns with this paper?

No

Have you any concerns about statistical analyses in this paper?

No

Recommendation?

Accept with minor revision (please list in comments)

Comments to the Author(s)

This is an interesting manuscript which reading I have enjoyed. The topic of this study is really worth research. There are only anecdotal reports about the importance of scent for roost recognition in bats and I appreciate effort conducted by the authors. If I have to point more serious concern, then it would be an extensive list of experiments made using a relatively small sample sizes. Instead, I would prefer one bullet-proof design with more individuals. Nevertheless, the results are consistent and do not question the overall message of the paper. Well, I understand the development of such experimental design and it can be useful if authors would argue a bit more about the need for different designs which they employed. Nevertheless, it is hard to follow descriptions of captive experiments in methods. Maybe, some general description for all versions accompanied with tabular specification for all six variations will help readers to follow differences.

Minor comments

P6, L26 - please specify which R-package was used with reference.

P8, L17 (and P13, L2) - put "boot" package to quotation marks and add reference for this package.

P9, L2-4 - You already report results here. Either do not report captive experiment one or move this information to results.

P14, L42-44 - Remove discussion from results. Use discussion for this information.

Review form: Reviewer 3 (Michael Schöner)

Is the manuscript scientifically sound in its present form?

Yes

Are the interpretations and conclusions justified by the results?

Yes

Is the language acceptable?

Yes

Do you have any ethical concerns with this paper?

No

Have you any concerns about statistical analyses in this paper?

No

Recommendation?

Accept with minor revision (please list in comments)

Comments to the Author(s)

In their manuscript „Do bats use scent to find roosts?“ Brown et al. conducted experiments with three different bat species in the wild and in captivity to evaluate the focal question.

The authors conducted eight different experiments and received very different and unclear results. It should be credited that the authors nevertheless seek to publish their results as they 1) give important information on what worked, where they did not get adequate results and how the study design should be changed in further studies. The question how bats do find new roosts still presents a largely unsolved puzzle and many authors suggested that olfaction might play a role but few ever tested this. 2) From a more general point of view and given today's publishing practice (and related the practice of how to apply for third-party funding) the paper is a best-practice example that science is not always that clear and should not be reduced to simple answers for complex questions.

It is further notable that the authors conducted experiments in the wild and in captivity. Finally, the choice to use bootstrap and permutation tests seems wise and as far as I can say the statistical analyses seem sound. However, some comments in one of the R scripts puzzled me a bit. Please see below.

I do have several questions and comments. Especially in the Materials & Methods section I feel that some improvements should be done so that the readers better understand what was done.

Title:

I think that the title is too generalized. Although the authors tested two bat species of the family *Vespertilionidae* and one of the family *Phyllostomidae*, it could not be answered if bats in general do or do not use scent to find roosts even if the results would have been clearer.

Abstract

Page 2, Line 50 – 52: Again, this statement is too generalized. Please change to “for the tested bat species.”

Introduction

I am missing some predictions here that better explain what exactly was tested in the experiments. E.g. “If bats use scent for finding roosts, they should approach and enter roosts stained with feces and urine more often than roosts without.” What is the authors’ prediction regarding the attractiveness of acoustic cues vs. scent?

Materials and Methods

Page 5, Line 41 – 46 (and similarly for the Ohio field experiment): The authors write: “We collected guano from four natural roosts of velvety freetailed bats that were not associated with the experiment. For each pair of artificial roosts, we selected guano from the natural roost that was located closest to the experiment site.”

I do not completely understand that: Can the authors really exclude that the closest natural roost from where feces had been collected derived from other individuals than those in the experiment? And if so, did the authors take into account that feces that had been collected from other individuals, maybe even another colony, might even have a deterring effect on conspecifics?

Can the authors estimate how old the feces were when they were originally collected in the roosts? How have they been preserved? For how long was a painted landing pad used before a new scent-mix was used?

Line 55 – 57: The authors write that they “broadcast echolocation and social calls for one hour immediately after sunset, when velvety-free tailed bats are active (Holland et al. 2011).” I wonder if this is the right time to attract bats to roosts? Wouldn’t they be more interested in going foraging at that time?

Line 58 – 60: The authors write that they “constructed playbacks from recordings of calls from flying velvety-free tailed bats (n=22) that were released from a platform and calling towards the microphone (...) from a previous study conducted at the same location”. Does that mean that the recorded individuals had been in captivity before? Can the authors exclude that especially the social calls were not used to warn conspecifics similar to distress calls? Can the authors exclude that the same individuals that had been recorded would not approach to the focal artificial roosts?

Page 11, Line 53 – 55: The authors write: “To create scent cues, we wetted two to three g of guano”. Was this also a mix of 2 to 3 g guano and 10 g water?

Results

Page 14, Line 46 – 60: I am a bit puzzled with the numbering of the experiments here. The authors write that captive experiment 5 was similarly designed as exp. 2 and 3. And then they mention exp. 4 but describe results of the y-maze. However, according to the description in the M & M section and fig. 2 the y-maze experiment should be exp. 5. I would suggest that for each experiment the authors should mention some key characteristics and the tested bat species to facilitate this section for the readers.

Discussion

Page 16, Field experiment: Is it possible to draw any conclusions from the field experiments at all? If the bats never entered the artificial roosts it might be possible that they either did not like that type of roost or at least that it was unfamiliar to them and that they would have needed more time getting to know them. Unlike in the captive experiments where the bats can only choose between an (initially) unfamiliar roost or hanging freely, the usually will have the option to find another familiar roost in the wild.

Page 17, Line 1 – 2: Is it possible that the inherent smell of the new roosts was so strong that it covered the guano scent and was deterring the bats?

Page 17, Captive experiments: The second and third interpretation in my mind sound promising. It may be that it is even a mixture of both interpretations: As the authors recognize themselves, scent alone might not be a honest cue because it should be difficult for the bats to decide if a roost actually is occupied or had been occupied shortly before. Echolocation/social calls, in contrast, can only be emitted if conspecifics/colony members are present. For group-dwelling animals this should be an important basis for decision-making. As mentioned above, especially in the wild individuals will have the chance to search for other roosts where they know to find group members by their calls. However, in captivity where there is no other than the offered roosts, scent is a better cue than nothing. Here it gets clear that using the same bats in different experiments was not too beneficial for the authors. Once the bats learnt that the scent did not lead them to other group members there was no reason anymore to rely on that cue.

Generally, it might be possible that there are different types of feces which serve different functions (e.g. scent to attract or to deter conspecifics) but that these different types of feces consist of different chemical compounds (i.e. they smell different) resulting in a different behavior of the conspecifics. For further studies it might be tested if bats that stick their feces to the entrance or outside of their roosts are attracted to roosts at least if the feces used in the experiments are 1) also taken from the entrance/outside of the roost and 2) derive from the same colony that is tested. For example, European Pipistrellus pipistrellus often roost in crevices of human buildings. These bats paste up the walls of the buildings with their feces. I would not know of any study that has ever tested if the feces fixed to the walls have an attracting function for colony members or a deterring function for non-group members or if the feces fixed to the walls show different chemical compounds than the feces inside the roosts.

R script Mollosus_FlyBy

As mentioned above there are some comments in this R script that have drawn my attention:

- “# load most of the data for the molossus field experiments” – why “most of the data” and not all connected data? Which data have been excluded and why?

- “# one week of raw data was lost so...

get the missing week of raw data from a previous dataset” – it should be mentioned in the M & M section that data got lost. How can missing data from a previous data set compensate for the missing data?

Best wishes

Michael Schöner

Decision letter (RSOS-201055.R0)

Dear Mrs Brown,

On behalf of the Editors, I am pleased to inform you that your Manuscript RSOS-201055 entitled "Do bats use scent to find roosts?" has been accepted for publication in Royal Society Open Science subject to minor revision in accordance with the referee suggestions. Please find the referees' comments at the end of this email.

The reviewers and handling editors have recommended publication, but also suggest some minor revisions to your manuscript. Therefore, I invite you to respond to the comments and revise your manuscript.

- Ethics statement

- Data accessibility

If you wish to submit your supporting data or code to Dryad (<http://datadryad.org/>), or modify your current submission to dryad, please use the following link:
<http://datadryad.org/submit?journalID=RSOS&manu=RSOS-201055>

- Competing interests

- Authors' contributions

- Acknowledgements

- Funding statement

Please ensure you have prepared your revision in accordance with the guidance at <https://royalsociety.org/journals/authors/author-guidelines/> -- please note that we cannot publish your manuscript without the end statements. We have included a screenshot example of

the end statements for reference. If you feel that a given heading is not relevant to your paper, please nevertheless include the heading and explicitly state that it is not relevant to your work.

Because the schedule for publication is very tight, it is a condition of publication that you submit the revised version of your manuscript before 08-Aug-2020. Please note that the revision deadline will expire at 00.00am on this date. If you do not think you will be able to meet this date please let me know immediately.

If your manuscript is newly submitted and subsequently accepted for publication, you will be asked to pay the article processing charge, unless you request a waiver and this is approved by Royal Society Publishing. You can find out more about the charges at <https://royalsocietypublishing.org/rsos/charges>. Should you have any queries, please contact openscience@royalsociety.org.

on behalf of Dr Claudia Wascher (Associate Editor) and Pete Smith (Subject Editor)
openscience@royalsociety.org

Associate Editor Comments to Author (Dr Claudia Wascher):

The authors investigate the use of olfactory cues for roost selection in bats. The reviewers find the research interesting and well presented. The reviewers highlight the value to publish negative results and complement that the authors properly discuss and acknowledge their findings. This is valuable for the field and future research. I agree with this evaluation. The reviewers have some suggestions regarding the presentation of the study, which should be addressed prior to publication.

Reviewer comments to Author:

Reviewer: 1
Comments to the Author(s)

This paper describes a series of field and lab experiments designed to detect the influence of olfactory cues on roost selection. This addresses some long-standing assumptions in the field of bat behavioral ecology that haven't been thoroughly tested. Despite the predominantly negative results, the experiments were logical, appropriate and systematically executed. The paper lacks any clear conclusions, other than demonstrating that there is not a clear or obvious scent bias when bats select temporary roosting sites. The principle benefit of the manuscript is that others will likely want to bypass these experiments and may find the negative results useful for building upon. It still seems probable that bats use olfactory cues for roost selection, mate attraction or territorial behaviors, but it might take more sophisticated, time-intensive and expensive efforts to quantify them.

Only main complaint is the way the experiments are introduced in the methods is very confusing and the presentation pattern isn't consistent going into the results. The 8 experiments outlined in methods are broken down into 8 headered sections varying by species, location, field or lab, and experimental conditions. There are double experiment 1s and 2s, which gets dizzying in the results, which only has two sections, field and lab, and bounces back and forth between different experimental numbers, species and locations. I would suggest consecutively numbering all the experiments 1-8, while still giving each an appropriate descriptive title in the header. Then in Results keep the two sections format but systematically go through and refer to experiment 1, 2, 3 etc. This would be easier for me to follow than trying to keep track of which experiment 1 (captive or field) is being discussed.

Minor Comments

Page 5, line 2: *Molossus molossus* field experiments would be a more appropriate title for this section, and then Page 7 line 2 *Eptesicus fuscus* field experiments. The species distinction seems more significant and relevant than where they were performed.

Page 7 line 43: Choosing “clipped” as the threshold is ok, but please provide details about what the threshold amplitude would be (i.e. >100 dB SPL). Presumably the recorders maxed out at 5 or 10 volts, and with the microphone sensitivity it should be straight forward to convert clipping to a loudness threshold. ON the other hand, Results-Field Experiments (page 13 lines 50+) indicates that only one *Eptesicus* call at 95% (i.e. below clipping) was detected, and none for *Molossus*. If NO calls caused clipping, then what is the point of defining this parameter at all?

Page 8 line 1: 10 minutes is a long time. Would this exclude counting bats travelling in pairs or groups? I’m guessing that if it mattered the paper would have said so, but would be worth knowing whether or not shortening this to 1 minute (or less) would have produced different results.

Page 8 line 39: typo, the otherwise same?

Page 8 line 51: Using the camcorder in the later experiments seems like a no-brainer, but here the “opportunistically scored after 1-24 hours” leaves room for criticism. Was there at least a regular schedule for checking the tubes over the first 24 hours, or was it assumed that once a bat chose a tube that they would continue to use just that tube?

Page 9 line 4: Typo “In captive experiment two experiment, we tested...”. Drop the second experiment.

Page 9 line 34, line 53, and elsewhere. This “Individual bat as the observational unit” statement is not a complete sentence and is repeated at the end of several sections.

Page 9, line 48, of captive experiment 3. At what time during the night were the bats assayed? It says every 10 minutes for four hours, which is fine, but when during the night did these four hours occur? Is there a problem here that the wild-caught bats would normally not want to roost during the night time, since they would normally be out feeding?

Page 9, Captive Experiment 4: How does covering a scented towel with a clean unscented towel control for echoacoustic cues? Need clarification on what the potential cues were.

Figures 3 and 4 are very interesting.

Discussion: While I generally agree that publishing negative results is useful where appropriate, the discussion goes overboard with this defense. Captive experiments 1 and 2 are messy and flawed and might fairly be considered unpublishable pilot experiments. The larger issue may be that short-term experiments like these may not be the best way to capture the role of olfaction in roost selection. As I’m sure the authors recognize, olfaction is likely just one of several cues getting integrated with seasonality, time of day/night, sex and reproductive status, stress and social factors etc. Still, collectively the eight experiments make a cohesive story that may save time for future investigators. Experiment 5 was very well done and would love to see that expanded upon.

Reviewer: 2

Comments to the Author(s)

This is an interesting manuscript which reading I have enjoyed. The topic of this study is really worth research. There are only anecdotal reports about the importance of scent for roost

recognition in bats and I appreciate effort conducted by the authors. If I have to point more serious concern, then it would be an extensive list of experiments made using a relatively small sample sizes. Instead, I would prefer one bullet-proof design with more individuals. Nevertheless, the results are consistent and do not question the overall message of the paper. Well, I understand the development of such experimental design and it can be useful if authors would argue a bit more about the need for different designs which they employed. Nevertheless, it is hard to follow descriptions of captive experiments in methods. Maybe, some general description for all versions accompanied with tabular specification for all six variations will help readers to follow differences.

Minor comments

P6, L26 - please specify which R-package was used with reference.

P8, L17 (and P13, L2) - put "boot" package to quotation marks and add reference for this package.

P9, L2-4 - You already report results here. Either do not report captive experiment one or move this information to results.

P14, L42-44 - Remove discussion from results. Use discussion for this information.

Reviewer: 3

Comments to the Author(s)

In their manuscript „Do bats use scent to find roosts?“ Brown et al. conducted experiments with three different bat species in the wild and in captivity to evaluate the focal question.

The authors conducted eight different experiments and received very different and unclear results. It should be credited that the authors nevertheless seek to publish their results as they 1) give important information on what worked, where they did not get adequate results and how the study design should be changed in further studies. The question how bats do find new roosts still presents a largely unsolved puzzle and many authors suggested that olfaction might play a role but few ever tested this. 2) From a more general point of view and given today's publishing practice (and related the practice of how to apply for third-party funding) the paper is a best-practice example that science is not always that clear and should not be reduced to simple answers for complex questions.

It is further notable that the authors conducted experiments in the wild and in captivity. Finally, the choice to use bootstrap and permutation tests seems wise and as far as I can say the statistical analyses seem sound. However, some comments in one of the R scripts puzzled me a bit. Please see below.

I do have several questions and comments. Especially in the Materials & Methods section I feel that some improvements should be done so that the readers better understand what was done.

Title:

I think that the title is too generalized. Although the authors tested two bat species of the family Vespertilionidae and one of the family Phyllostomidae, it could not be answered if bats in general do or do not use scent to find roosts even if the results would have been clearer.

Abstract

Page 2, Line 50 – 52: Again, this statement is too generalized. Please change to “for the tested bat species.”

Introduction

I am missing some predictions here that better explain what exactly was tested in the experiments. E.g. “If bats use scent for finding roosts, they should approach and enter roosts stained with feces and urine more often than roosts without.” What is the authors' prediction regarding the attractiveness of acoustic cues vs. scent?

Materials and Methods

Page 5, Line 41 – 46 (and similarly for the Ohio field experiment): The authors write: “We collected guano from four natural roosts of velvety freetailed bats that were not associated with the experiment. For each pair of artificial roosts, we selected guano from the natural roost that was located closest to the experiment site.”

I do not completely understand that: Can the authors really exclude that the closest natural roost from where feces had been collected derived from other individuals than those in the experiment? And if so, did the authors take into account that feces that had been collected from other individuals, maybe even another colony, might even have a deterring effect on conspecifics?

Can the authors estimate how old the feces were when they were originally collected in the roosts? How have they been preserved? For how long was a painted landing pad used before a new scent-mix was used?

Line 55 – 57: The authors write that they “broadcast echolocation and social calls for one hour immediately after sunset, when velvety-free tailed bats are active (Holland et al. 2011).” I wonder if this is the right time to attract bats to roosts? Wouldn't they be more interested in going foraging at that time?

Line 58 – 60: The authors write that they “constructed playbacks from recordings of calls from flying velvety-free tailed bats (n=22) that were released from a platform and calling towards the microphone (...) from a previous study conducted at the same location”. Does that mean that the recorded individuals had been in captivity before? Can the authors exclude that especially the social calls were not used to warn conspecifics similar to distress calls? Can the authors exclude that the same individuals that had been recorded would not approach to the focal artificial roosts?

Page 11, Line 53 – 55: The authors write: “To create scent cues, we wetted two to three g of guano”. Was this also a mix of 2 to 3 g guano and 10 g water?

Results

Page 14, Line 46 – 60: I am a bit puzzled with the numbering of the experiments here. The authors write that captive experiment 5 was similarly designed as exp. 2 and 3. And then they mention exp. 4 but describe results of the y-maze. However, according to the description in the M & M section and fig. 2 the y-maze experiment should be exp. 5. I would suggest that for each experiment the authors should mention some key characteristics and the tested bat species to facilitate this section for the readers.

Discussion

Page 16, Field experiment: Is it possible to draw any conclusions from the field experiments at all? If the bats never entered the artificial roosts it might be possible that they either did not like that type of roost or at least that it was unfamiliar to them and that they would have needed more time getting to know them. Unlike in the captive experiments where the bats can only choose between an (initially) unfamiliar roost or hanging freely, the usually will have the option to find another familiar roost in the wild.

Page 17, Line 1 – 2: Is it possible that the inherent smell of the new roosts was so strong that it covered the guano scent and was deterring the bats?

Page 17, Captive experiments: The second and third interpretation in my mind sound promising. It may be that it is even a mixture of both interpretations: As the authors recognize themselves, scent alone might not be a honest cue because it should be difficult for the bats to decide if a roost actually is occupied or had been occupied shortly before. Echolocation/social calls, in contrast, can only be emitted if conspecifics/colony members are present. For group-dwelling animals this should be an important basis for decision-making. As mentioned above, especially in the wild

individuals will have the chance to search for other roosts where they know to find group members by their calls. However, in captivity where there is no other than the offered roosts, scent is a better cue than nothing. Here it gets clear that using the same bats in different experiments was not too beneficial for the authors. Once the bats learnt that the scent did not lead them to other group members there was no reason anymore to rely on that cue.

Generally, it might be possible that there are different types of feces which serve different functions (e.g. scent to attract or to deter conspecifics) but that these different types of feces consist of different chemical compounds (i.e. they smell different) resulting in a different behavior of the conspecifics. For further studies it might be tested if bats that stick their feces to the entrance or outside of their roosts are attracted to roosts at least if the feces used in the experiments are 1) also taken from the entrance/outside of the roost and 2) derive from the same colony that is tested. For example, European Pipistrellus pipistrellus often roost in crevices of human buildings. These bats paste up the walls of the buildings with their feces. I would not know of any study that has ever tested if the feces fixed to the walls have an attracting function for colony members or a deterring function for non-group members or if the feces fixed to the walls show different chemical compounds than the feces inside the roosts.

R script Mollosus_FlyBy

As mentioned above there are some comments in this R script that have drawn my attention:

- "# load most of the data for the molossus field experiments" - why "most of the data" and not all connected data? Which data have been excluded and why?

- "# one week of raw data was lost so...

get the missing week of raw data from a previous dataset" - it should be mentioned in the M & M section that data got lost. How can missing data from a previous data set compensate for the missing data?

Best wishes

Michael Schöner

Author's Response to Decision Letter for (RSOS-201055.R0)

See Appendix A.

Decision letter (RSOS-201055.R1)

Dear Mrs Brown,

It is a pleasure to accept your manuscript entitled "Do bats use guano and urine stains to find new roosts? Tests with three group-living species" in its current form for publication in Royal Society Open Science.

You can expect to receive a proof of your article in the near future. Please contact the editorial office (openscience_proofs@royalsociety.org) and the production office (openscience@royalsociety.org) to let us know if you are likely to be away from e-mail contact -- if

you are going to be away, please nominate a co-author (if available) to manage the proofing process, and ensure they are copied into your email to the journal.

on behalf of Dr Claudia Wascher (Associate Editor) and Pete Smith (Subject Editor)
openscience@royalsociety.org

Do bats use ~~seent~~guano and urine stains to find new roosts? Tests with three group-living
bats

**Bridget K. G. Brown^{a*}, Lauren Leffer^c, Yesenia Valverde^b, Nia Toshkova^{bd},**
**Jessica Nystrom^a, Rachel A. Page^b, Gerald G. Carter^{ab}**

5 ^a*†Department of Evolution, Ecology, and Organismal Biology, The Ohio State*
*University, 318 W. 12th Avenue, Columbus, Ohio, USA*

7 ^b*Smithsonian Tropical Research Institute, Apartado 0843-03092, Balboa, Ancón,*
*Republic of Panama*

9 ^c*University of Maryland, 4094 Campus Drive, College Park, Maryland, USA*

10 ^d*National Museum of Natural History at the Bulgarian Academy of Science, 1000*
11 *Sofia Center, Sofia, Bulgaria*

12 **Keywords:** bats; *Desmodus*; *Molossus*; *Eptesicus*; roosting ecology; sensory ecology
13

**Abstract**

Many animals use social cues to find refuges. Bats can find roosts using the echolocation and
social calls of conspecifics, but they might also use scent cues, a possibility which is less studied. The
entrances of bat roosts are often marked by guano and urine, ~~which could provide~~providing possible
scent cues. We conducted eight experiments to test whether bats use the scent of guano and urine to
find potential roosts. In field experiments, we tested if ~~velvety free-tailed bats~~ (*Molossus molossus*
(Velvety Free-tailed Bats)) in Panama and ~~big-brown bats~~ (*Eptesicus fuscus* (Big Brown Bats)) in Ohio
would ~~more likely~~ investigate ~~paired~~ artificial roost boxes that were ~~either~~ scented with guano and
urine ~~or more often than a paired~~ unscented control. We did not detect any difference in flights near
the scented versus unscented roosts, and we detected only one entrance into ~~an~~any artificial roost
(scented). In six captive experiments, we tested for attraction of ~~common vampire bats~~ (*Desmodus*
*rotundus* (Common Vampire Bats)) and ~~velvety free-tailed bats~~ *Molossus molossus* to areas scented
with guano and urine, under several conditions. Results were mixed, but overall suggested that the

*Author for correspondence (brown.6531@buckeyemail.osu.edu).

†Present address: Department of Evolution, Ecology, and Organismal Biology, 318 W. 12th Avenue,
Columbus, Ohio, USA

scent of guano and urine does not act as a strong lure for ~~bats~~ the tested bat species. We suggest that
further tests of olfaction-based roost choice in bats should manipulate existing scent cues on familiar
roosts.

**Introduction**

A key benefit of living in groups is that individuals can gain social information about
resources, such as food or roosts (Bonnie and Earley 2007). For bats, potential roosts are often
limited, and social information from conspecifics can ~~aid help~~ aid help bats ~~in finding find~~ find new roosts (Kerth
2008). In some species and at some times of year, roost-~~searching~~ searching behaviour ~~can occur occurs~~
frequently. For example, disc-winged bats roost in young furled leaves and move to new roosts daily
(Chaverri 2010). To find unfamiliar roosts, bats can rely on the echolocation and social calls
produced by conspecifics (Chaverri 2010; Furmankiewicz et al. 2011; Ruczyński and Kalko 2007;
Ruczyński et al. 2009; Schöner et al. 2010). However, it remains unclear to what extent bats find and
select roosts using other social cues, such as chemical cues.

There are several reasons why we might expect bats to use scent to locate roosts. Bats use
scent cues for mate selection (Chaverri et al. 2018), foraging (Theis et al. 2016), and social
recognition (Bloss et al. 2002; De Fanis and Jones 1995; Safi and Kerth 2003; Voigt and von
Helversen 1999). Compared to high-frequency bat calls, chemical cues can potentially travel farther
distances, especially in cluttered forests, and ~~will last for longer periods of they remain over~~ will last for longer periods of they remain over time
(Martens 1980). Predators and dogs can use guano and urine cues to locate bat roosts (Chambers et
al. 2015; Delpietro et al. 2017; Threlfall et al. 2013). In at least one bat species (*Hipposideros*
*speoris*), males appeared to use urine to relocate roosting spots and establish territory boundaries
within a cave (Selvanayagam and Marimuthu 1984). Some anecdotal reports suggest that guano
stains can attract bats to a bat house, though others have argued that such stains have no effect
(Murphy 1993; Ober 2014). If bats can be attracted to artificial roosts using a chemical lure, this
could have important implications for bat conservation and management.

~~We~~ In this study, we focused on three common group-living bat species, ~~velvety free-tailed~~
~~bats~~ (*Molossus molossus*), ~~big brown bats~~ (Velvety Free-tailed Bats), *Eptesicus fuscus* (Big Brown

~~Bats~~), and ~~common vampire bats~~ (*Desmodus rotundus* (Common Vampire Bats)) for ~~three~~ several
reasons. First, they form stable social networks with preferred associations (Gager et al. 2016;
Metheny et al. 2008; Wilkinson 1990; Willis and Brigham 2004). ~~Big brown bats~~ *Eptesicus fuscus*
and ~~common vampire bats~~ *Desmodus rotundus* also frequently switch roosts or roosting locations
(Wilkinson 1985; Willis and Brigham 2004).

Second, these species ~~also~~ live within communal roosts that are often stained with guano and
urine. ~~Vampire bats~~ *Desmodus*, in particular, often live in roosts that are uniquely odorous because
their diet of blood leads to copious urine and tar-like guano such that their pungent roosts can be
detected at a distance by dogs or humans (Chambers et al. 2015; Delpietro et al. 2017). All three
species will also roost in anthropogenic structures, showing some plasticity in their roost choices.

Third, our study species are likely to use olfaction in social interactions. ~~The neotropical~~
~~common vampire bat~~ *D. rotundus* is attracted to the scent of prey (Bahlman and Kelt 2007) ~~and~~,
appears to use scent to aid in social recognition (Carter and Wilkinson 2013). ~~Vampire bats possess,~~
~~and possesses~~ an intact vomeronasal system and roughly twice as many intact vomeronasal type-1
receptor genes as other bats (Yohe et al., 2019). ~~The velvety free-tailed bat~~ *Molossus molossus* and
~~big brown bat~~ *Eptesicus fuscus* also have a vomeronasal organ ~~suggesting some~~ indicating a capacity
to detect social signals in pheromones, but they lack a vomeronasal epithelial tube and an accessory
olfactory bulb suggesting a less developed olfactory system than other mammals (Wible and
Bhatnagar 1996). Despite this, ~~big brown bats~~ *Eptesicus fuscus* can discriminate between familiar and
non-familiar conspecifics using scent (Bloss et al. 2002). Though olfaction is not well-studied in
~~velvety free-tailed bats~~ *Molossus molossus*, the related Brazilian free-tailed bat, *Tadarida brasiliensis*,
uses conspecific scent to locate and identify its pups, roostmates, and roosting sites (Englert and
Greene 2009; Gustin and McCracken 1987).

We conducted eight experiments to test whether bats are more likely to visit roosting sites
when they are stained with guano and urine: a field ~~and a captive~~ experiment with ~~velvety free-tailed~~
~~bats~~ *Molossus molossus*, a field experiment with ~~big brown bats,~~ ~~and~~ *Eptesicus fuscus*, five captive
experiments with ~~common vampire bats~~ *Desmodus rotundus*, ~~and a captive experiment with~~ *Molossus*
*molossus*. In two of these experiments, we also presented playbacks of bat calls to compare the

[revised manuscript text omitted]

To count echolocation calls and possible visits of ~~big brown bats~~ *Eptesicus fuscus*, we used the
same procedures described above for the ~~Panama~~ *Molossus molossus* field experiment. To target ~~big~~
~~brown bat~~ *Eptesicus fuscus* calls, we selected calls across the whole night that had peak frequencies of
25-40 kHz and durations of 5-20 ms (Surlykke et al. 2009). We did not conduct any playback
experiments in Ohio.

*Statistical analysis for field experiments*

~~We compared three measures of activity within each pair of boxes.~~ First, to check for clear
roost ‘visits’ by bats we looked for clipped calls (i.e. those that exceeded the maximum possible
amplitude) in each of the paired boxes, ~~which~~. ~~We knew that clipped calls~~ would indicate that a bat
was on the landing pad and calling directly into the microphone inside that box. ~~This method of~~
~~detecting visits was confirmed with~~ based on recording three captive ~~big brown bats~~ *Eptesicus fuscus*
~~(from the Ohio Wildlife Center, which would produce many clipped calls when echolocating into the~~
~~box after being) that were~~ placed onto the landing pad of ~~athe~~ roost box (n=3).

Second, to investigate whether the highest echolocation intensity calls (i.e. closest
approaches) were more likely to be recorded at the scented box than the unscented box, we manually
inspected the spectrograms of bat calls with an amplitude over 5% of the maximum possible
amplitude. To ensure calls were independent bat approaches, we only counted the loudest calls
separated by 10 min of each other. This is a fairly conservative approach, but our conclusions do not
change when instead we used a separation time of one minute or when not excluding any calls. We
then conducted a binomial test to test whether the loudest bat calls were more likely to be recorded at
the scented box.

Third, to assess bat activity near each box, we counted the number of possible bat
echolocation calls at the scented box and the unscented box within each night and site. We then
measured the scent bias as the difference between the number of calls at each box (scented –
unscented) divided by the total number of calls at both boxes (scented + unscented) for each
independent site. To calculate a nonparametric 95% confidence interval (CI) around the bias for each
experiment, we bootstrapped the mean scent biases using the “boot” package (Canty & Ripley 2020)
in R (5000 iterations, “boot” package, percentile method).

**Results for Field Experiments**

In Ohio, we did not detect any clipped calls (100% maximum amplitude), but we detected one
clear case of an approach: an extreme outlier in call intensity (i.e. 99.75% of the maximum amplitude
compared to the next loudest call, which was 58% of the amplitude and 508,896 other values which
ranged from 3.7 to 27% of the maximum amplitude). This case suggested that only one *Eptesicus*
*fuscus* echolocated directly into a scented roost box. We did not detect a difference in the total
number of calls at the scented and unscented boxes (two sites, five weeks, permuted paired t-test:
$t=0.58$, $p=0.5$). Only 11 of the 27 closest bat passes (i.e. loudest call bouts) were at the scented box
(binomial test: $p=0.44$).

In Panama, we did not detect any evidence for visits from *Molossus molossus* at either box.
We did not detect more calls at the scented boxes; in fact, the sample mean was trending to be higher
at the unscented control box (six sites, six weeks, permuted paired t-test: $t=-1.86$, $p=0.08$). Only eight
of the 24 closest bat passes (loudest call bouts) were at the scented box (binomial test: $p=0.15$). We
also did not detect more calls during the playback hours (432 calls per hour, 95% CI: [274, 583]) than
during the corresponding non-playback hours (505 calls per hour [311, 677]).

**Methods for Captive Experiments**

*Captive Experiment One: ~~Vampire bats~~ *Desmodus rotundus* choosing dark locations to hide from*
*light*

We conducted five experiments with captive vampire bats (*Desmodus rotundus*), each with
 different observers, and each improving on the methods of the previous test. In the first experiment,
 we repeatedly tested five captive ~~adult common vampire bats~~ adults (four females and one male,
 originally from Chicago Brookfield Zoo), by presenting them with a scented and an unscented
 refuge, which consisted of rolls of plastic mesh covered in paper and black plastic (Figure 2a). The
 scented tube contained a mesh bag at the top with a paper towel and bedding soiled with guano and
 urine from the bat's home enclosure. The unscented tube ~~was the otherwise same, except~~
 ~~with~~ contained unsoiled paper towels and bedding and was otherwise the same. To induce bats into
 choosing to roost inside one of the tubes, we used a dim light to ensure that the tubes were the darkest
 location to roost. We tested each bat individually and repeatedly in three small cages (0.3 x 0.3 x 0.4
 11 m) and one flight cage (1.7 x 2.1 x 2.3 m). In the flight cage, we also conducted tests with one
 scented and three unscented tubes. Bats did not immediately enter the roost tubes, ~~so their locations~~
 ~~were opportunistically scored after 1-24 h until the bat was inside a tube, and would often hide in a~~
 ~~corner of the cage before moving into the darker roost tube. We therefore repeatedly checked their~~
 ~~location at the next convenient time until the bat was inside a tube (1-24 h). Due to this haphazard~~
 ~~sampling, we could not measure latency or be sure whether this choice was their first choice, but we~~
 ~~suspect that, once a bat selected and hid inside a dark tube, it probably did not leave that refuge in the~~
 ~~presence of ambient light~~. If the bat did not choose either tube, the cage was checked at least one h
 later. After all five bats made a choice, we swapped the bats into new test cages and swapped the
 locations of the scented tubes within each cage. Each bat made two to six roost choices in different
 test cages (n=19 choices). Individual bat as was the observational unit.

 *Captive Experiment Two: ~~Vampire bats~~ *Desmodus rotundus* from zoo choosing surfaces from which*
 *to hang*

~~In Captive Experiment One, bats often did not enter the artificial roosts, perhaps due to~~
 ~~neophobia, or perhaps due to our limited sample size (n=5).~~ In Captive Experiment Two ~~experiment,~~
 we tested a greater number of bats (n=20, ~~15 female and five male adult vampire bats~~) and than in
 Captive Experiment One (n=5). We also used a test chamber (~~33 x 33 x 19 cm, clear plastic tank~~) that

was familiar to the bats to remove possible neophobia that may have affected Captive Experiment
One. The bats tested were from a long-term captive colony at the Organization for Bat Conservation
in Bloomfield Hills, Michigan (described in Carter and Wilkinson 2013). To test for a preference to
hang near or on a scented surface, we attached a clean paper towel sheet to one side of the tank and a
paper towel sheet stained with urine and guano on the other side. The guano and urine were collected
daily from the floor below the captive colony. The bat could hang from the scented side, the
unscented side, or the ceiling, but it could not hang from the remaining two clear sides of the test
chamber that were smooth (Figure 2b). We left an individual bat in the chamber in the dark, and then
opportunistically sampled the position of the bat after a period of one to five h. If the bat was on the
floor or ceiling of the chamber, we took another sample at least one h later. We swapped the location
of the scented towel and cleaned the chamber between each tested bat. Each bat made one to nine
choices (n=48 choices). Individual bat aswas the observational unit.

*Captive Experiment Three: ~~Vampire bats~~Desmodus rotundus from Panama choosing surfaces from*
*which to hang*

For Captive Experiment Three, we replicated Captive Experiment Two using 22 vampire
batsDesmodus rotundus (14 wild-caught adult females, four captive-born males, four captive-born
females) in Panama, but here we recorded each individual bat using infrared light and a Sony
Nightshot camcorder (hereon ‘infrared video’). We then sampled the roosting position of the bat
every 10 min for four h within the chamber: (from midnight to 0400h). During this time, bats in the
colony are likely to be active and at rest. Choices were defined as a bat touching the scented or
unscented side. Each bat made one to three choices (n=39 choices). Individual bat aswas the
observational unit.

*Captive Experiment Four: ~~Vampire bats~~Desmodus rotundus choosing between spaces in divided*
*chamber*

In Captive Experiment Four, we attempted to remove possible visual and echoacoustic
differences between cues caused by guano on the scented and unscented sides of the chamber paper

towel (e.g. a darker surface with more texture) by covering the scented paper towel with two clean
paper ~~towel,towels~~ and placed three clean paper towels in a similar manner on the other side. We
tested 33 different adult ~~vampire bats~~*Desmodus rotundus* (28 females and five males) captured in
Panama. Bats were placed in a test chamber (40 cm x 56 cm x 40 cm, glass) and could choose to
hang on one of two sides divided by a cardboard divider covered in plastic (Figure 2c). The scent
cues were five g of guano and urine collected every one to three days from a captive vampire bat
colony. Between each trial, we removed the paper towels, cleaned the entire chamber, and swapped
the location of the scent treatment. To avoid potential biases caused by airflow or lighting, we also
rotated the position of the test chamber inside the room between trials. We recorded the trials using
infrared video. Bats were held in a cloth bag, which we placed in the center of the chamber below the
divider facing towards the back wall, allowing the bat to crawl out of the bag on their own. We tested
each bat one to four times in 91 trials (48 trials that were one h and 43 trials that were four h).

Individual bat aswas the observational unit.

[revised manuscript text omitted]

Field Experiments

~~In Ohio, we detected only one clear case of a visit: a big brown bat echolocated directly into a scented roost box (one call with an intensity greater than 95% of the maximum possible amplitude). We did not detect a difference in the total number of calls at the scented and unscented boxes (two sites, five weeks, permuted paired t test: $t=0.58$, $p=0.5$). Only 11 of the 27 closest bat passes (i.e. loudest call bouts) were at the scented box (binomial test: $p=0.44$).~~

~~In Panama, we did not detect any evidence for visits from velvety free-tailed bats at either box. We did not detect more calls at the scented boxes; in fact, the sample mean was trending to be higher at the unscented control box (six sites, six weeks, permuted paired t test: $t=-1.86$, $p=0.08$). Only eight of the 24 closest bat passes (loudest call bouts) were at the scented box (binomial test: $p=0.15$). We also did not detect more calls during the playback hours (432 calls per hour, 95% CI: [274, 583]) than during the corresponding non-playback hours (505 calls per hour [311, 677]).~~

for Captive Experiments

Overall, we did not detect a consistent and strong bias towards roosting in locations scented with guano and urine for the five experiments with ~~vampire bats~~ *Desmodus rotundus* (weighted mean=0.18, 95% CI: [-0.06, 0.38]) or for all six captive experiments (conservative weighted mean bias=0.15, 95% CI: [-0.06 0.32]; non-conservative weighted mean bias=0.15, 95% CI: [0.02,0.29]).

Results from all captive experiments are shown in Figure 3. In Captive Experiment One, (*Desmodus rotundus* choosing dark locations to hide from light), the five ~~vampire~~ bats often chose not to enter either a scented or an unscented roost tube. ~~When, but when~~ they did choose to enter a roost tube, all five bats chose scented tubes more often. (Figure 3). Bats chose to roost in the scented tube over the unscented tube in nine of 10 choices in the small cages with two tubes, and in five of eight choices in the larger flight cage trials with one scented roost and three unscented roosts. If we assume these choices were unbiased and independent, the binomial probability of seeing these outcomes by chance is only 0.02 and 0.03, respectively. ~~Unfortunately, however, these choices were not independent, because the same five bats were tested multiple times. We discuss what this might mean in the discussion section below.~~

1 In Captive Experiment Two, bats were more likely to hang on or at the surface stained with
2 guano and urine than on the unscented surface; however, when we ~~replicated~~attempted to replicate
this test ~~in~~with a group of wild-caught bats (Captive Experiment Three,~~),~~ we no longer detected such
a clear bias. In Captive Experiment ~~Five, which was similar in design~~Four, we removed possible
visual and echoacoustic cues and used an altered setup, and we detected no evidence for a scent bias.
When we repeated Captive Experiment Four with *Molossus molossus* instead of *Desmodus rotundus*
(Captive Experiment Six), we did not see a bias toward either the scented or unscented side.

~~In~~Finally, in Captive Experiment ~~Four~~Five, we compared an immediate attraction of wild-

[revised manuscript text omitted]

11 **Competing Interests**

12 We have no competing interests.

**Authors' Contributions**

BKGB, GGC, LL, YV, NT, and JN collected the data. BKGB and GGC analysed the data. BKGB
and GGC drafted the manuscript, and all authors critically revised the manuscript. GGC and RAP
supervised the work.

**Funding Statement**

GGC was supported by student research grants from the Animal Behaviour Society and the American
Society of Mammalogists, a Doctoral Dissertation Improvement Grant from the National Science
Foundation (1311336). LL was supported in part by a grant to the University of Maryland from the
Howard Hughes Medical Institute through the Science Education Program. BKGB was supported by
student research grants from Sigma Xi Grants-in-Aid of Research and Ohio State's Critical
Difference for Women Professional Development Grant.

Acknowledgments

We thank R. Mies, A. Felk, and J. Fabian for support and access to captive bat colonies and E. Jacobs
for help with data collection. We thank the Smithsonian Tropical Research Institute for logistical
support in Panama, in particular, G. Cohen, M. Dixon, A. Heckley, C. Santiago, and S. Stockmaier.
We also would like to express our gratitude towards J. Stancourt and M. Little for their assistance
with data collection on the Ohio Field Experiment. We would like to provide further
acknowledgement to the following people for helping to procure guano and roosting locations for the
field experiments: The Ohio Wildlife Center, Columbus and Franklin County Metro Parks Staff
(particularly C. Morrow), L. Cooper at NEOMED, and J. Kohles in Panama. Finally, we would like
to thank J. Kohles for also providing playback calls for field experiments with ~~velvety free-tailed~~
~~bats~~ *Molossus molossus*.

~~Ethical Statement~~

~~This work was approved by the University of Maryland Institutional Animal Care and Use~~
~~Committee (Protocol R-10-63), the Smithsonian Tropical Research Institute Animal Care and Use~~
~~Committee (#2015-0915-2018-A9, #2019-0501-2022), the Panamanian Ministry of the Environment~~
~~(#SE/A-76-16, #SE/A-36-19), and the Columbus and Franklin County Metro Parks through an access~~
~~permit.~~

~~Funding Statement~~

~~GGC was supported by student research grants from the Animal Behaviour Society and the American~~
~~Society of Mammalogists, a Doctoral Dissertation Improvement Grant from the National Science~~
~~Foundation (1311336). LL was supported in part by a grant to the University of Maryland from the~~
~~Howard Hughes Medical Institute through the Science Education Program. BKGB was supported by~~
~~student research grants from Sigma Xi Grants in Aid of Research and Ohio State's Critical~~
~~Difference for Women Professional Development Grant.~~

~~Data Accessibility~~

~~Data and R code are available on Figshare [link to be included]. These are temporary links for the reviewers [https://www.dropbox.com/sh/5j91mfdlmbwge3/AADcXZ2Exv9DTWiZTgAqk-d-a?dl=0; https://www.dropbox.com/sh/uvd3qf67j4vt5bp/AABXydOxed1joq2eFyz_KHwRa?dl=0].~~

~~Competing Interests~~

~~We have no competing interests.~~

~~Authors' Contributions~~

~~BKGB, GGC, LL, YV, NT, and JN collected the data. BKGB and GGC analysed the data. BKGB and GGC drafted the manuscript, and all authors critically revised the manuscript. GGC and RAP supervised the work.~~

-

References

1. Arnold, B. D., & Wilkinson, G. S. 2011. *Individual specific contact calls of pallid bats (Antrozous pallidus) attract conspecifics at roosting sites.* Behavioral Ecology and Sociobiology, **65**(8), 1581–1593.
2. Bahlman, J. W., & Kelt, D. 2007. *Use of olfaction during prey location by the common vampire bat (Desmodus rotundus).* Biotropica, **39**(1), 147–149.
3. Barclay, R. M. R., Faure, P. A., & Farr, D. R. 1988. *Roosting behavior and roost selection by migrating silver-haired bats (Lasionycteris noctivagans).* Journal of Mammalogy, **69**(4), 821–825. <https://doi.org/10.2307/1381676>
4. Barclay, R. M. R. 1982. *Interindividual use of echolocation calls: Eavesdropping by bats.* Behavioral Ecology and Sociobiology, **10**(4), 271–275. (<https://doi.org/10.1007/BF0085302816>)
5. Beynon, R. J., & Hurst, J. L. 2004. *Urinary proteins and the modulation of chemical scents in mice and rats.* Peptides, **25**(9), 1553–1563.
6. Bloss, J., Acree, T. E., Bloss, J. M., Hood, W. R., & Kunz, T. H. 2002. *Potential use of chemical cues for colony-mate recognition in the big brown bat, Eptesicus fuscus.* Journal of Chemical Ecology, **28**(4), 819–834. (<https://doi.org/10.1023/A:10115296928423>)
7. Bonnie, K.E., & Earley, R. L. 2007. *Expanding the scope for social information use.* Animal Behaviour, **74**(2), 171–181.
8. Bouchard, S. 2001. *Sex discrimination and roostmate recognition by olfactory cues in the African bats, Mops condylurus and Chaerephon pumilus (Chiroptera: Molossidae).* Journal of Zoology, **254**(1), 109–117. (<https://doi.org/10.1017/S09512836901000607>)
9. Brigham, R. M., Vonhof, M. J., Barclay, R. M. R., & Gwilliam, J. C. 1997. *Roosting behavior and roost site preferences of forest-dwelling California bats (Myotis californicus).* Journal of Mammalogy, **78**(4), 1231–1239. (<https://doi.org/10.2307/1383066>)
10. Canty, A., & Ripley, B.D. 2020. boot: Bootstrap R (S-Plus) Functions. R package version 1.3-25. 10.11. Carter, G. G., & Wilkinson, G. S. 2013. *Food sharing in vampire bats: reciprocal help predicts donations more than relatedness or harassment.* Proceedings of the Royal Society B: Biological Sciences, **280**(1753), 20122573–20122573. (<https://doi.org/10.1098/rspb.2012.2573>)
11. 11.12. Carter, G., Schoeppler, D., Manthey, M., Knörnschild, M., & Denzinger, A. 2015. *Distress calls of a fast-flying bat (Molossus molossus) provoke inspection flights but not cooperating mobbing.* PLOS ONE, **10**(9), e0136146. (<https://doi.org/10.1371/journal.pone.0136146>)
12. 12.13. Carter, Gerald G., & Wilkinson, G. S. 2016. *Common vampire bat contact calls attract past food-sharing partners.* Animal Behaviour, **116**, 45–51. (<https://doi.org/10.1016/j.anbehav.2016.03.005>)
13. 13.14. Caspers, B. A., Schroeder, F. C., Franke, S., Streich, W. J., & Voigt, C. C. 2009. *Odour-based species recognition in two sympatric species of sac-winged bats (Saccopteryx bilineata, S. leptura): Combining chemical analyses, behavioural observations and odour preference tests.* Behavioral Ecology and Sociobiology, **63**(5), 741–749. (<https://doi.org/10.1007/s00265-009-0708-7>)
14. 14.15. Chambers, C. L., Vojta, C. D., Mering, E. D., & Davenport, B. 2015. *Efficacy of scent-detection dogs for locating bat roosts in trees and snags: Using*

- *detection dogs to locate bat* 56 *benefit from sensory ecology* 110 27-28. Hecker, N., Lächele,
*roosts*. Wildlife Society 57 Nature Ecology and 111 U., Stuckas, H., Giere, P., &
*Bulletin*, **39**(4), 780–787. 58 Evolution, **4**, 502–511. 112 Hiller, M. 2019. *Convergent*
(<https://doi.org/10.1002/wsb>. 59 21-22. Englert, A. C., & 113 *vomeroneasal system*
598) 60 Greene, M. J. 2009. 114 *reduction in mammals*
15-16. Chaverri, G. 2010. 61 *Chemically-mediated* 115 *coincides with convergent*
*Comparative social network* 62 *roostmate recognition and* 116 *losses of calcium signaling*
*analysis in a leaf-roosting* 63 *roost selection by Brazilian* 117 *and odorant-degrading*
*bat*. Behavioral Ecology and 64 *free-tailed Bats (*Tadarida** 118 *genes*. Molecular Ecology,
Sociobiology, **64**(10), 1619– 65 *brasiliensis*). PLOS ONE, 119 **28**(16), 3656–3668.
1630. 66 **4**(11), e7781. 120 (<https://doi.org/10.1111/mec>.
16-17. Chaverri, G., 67 (<https://doi.org/10.1371/journal>
Ancillotto, L., & Russo, D. 68 al.pone.0007781) 121 15180)
2018. *Social communication* 69 22-23. Finn, L. S. 1997. *Bat* 122 28-29. Ho, J., Tumkaya, T.,
*in bats*. Biological Reviews, 70 *house use in central Florida,* 123 Aryal, S., Choi, H., &
**93**(4), 1938–1954. 71 *with emphasis on *Nycticeius** 124 Claridge-Chang, A. 2019.
(<https://doi.org/10.1111/brv>. 72 *humeralis and *Tadarida** 125 *Moving beyond P values:*
2427) 73 *brasiliensis cynocephala*. 126 *data analysis with estimation*
17-18. Chaverri, G., Gillam, 74 PhD thesis, University of 127 *graphics*. Nature Methods,
E. H., & Vonhof, M. J. 2010. 75 Central Florida, Orlando, 128 **16**(7), 565–566.
Social calls used by a leaf- 76 Florida. 129 (<https://doi.org/10.1038/s415>
roosting bat to signal 77 23-24. Furmankiewicz, J., 130 92-019-0470-3)
location. Biology Letters, **6**, 78 Ruczyński, I., Urban, R., & 131 29-30. Holland, R. A.,
441–444. 79 Jones, G. 2011. *Social calls* 132 Meyer, C. F. J., Kalko, E. K.
18-19. De Fanis, E., & Jones, 80 *provide tree-dwelling bats* 133 V., Kays, R., & Wikelski, M.
G. 1995. *Post-natal growth,* 81 *with information about the* 134 2011. *Emergence time and*
*mother-young interactions* 82 *location of conspecifics at* 135 *foraging activity in Pallas’*
*and development of* 83 *roosts*. Ethology, **117**(6), 136 *mastiff bat, *Molossus**
*vocalizations in the* 84 480–489. 137 *molossus (Chiroptera:*
*vespertilionid bat *Plecotus** 85 (<https://doi.org/10.1111/j>. 138 *Molossidae) in relation to*
*auritus*. Journal of Zoology 86 9-0310.2011.01897.x) 139 *sunset/sunrise and phase of*
(London), **235**, 85–97. 87 24-25. Gager, Y., Gimenez, 140 *the moon*. Acta
19-20. Delpietro, H. A., 88 O., O’Mara, M. T., & 141 Chiropterologica, **13**(2), 399–
Russo, R. G., Carter, G. G., 89 Dechmann, D. K. 2016. 142 404.
Lord, R. D., & Delpietro, G. 90 *Group size, survival and* 143 (<https://doi.org/10.3161/1508>
36 L. 2017. *Reproductive* 91 *surprisingly short lifespan in* 144 11011X624875)
*seasonality, sex ratio and* 92 *socially foraging bats*. BMC 145 30-31. Höller, P., & Schmidt,
*philopatry in Argentina’s* 93 Ecology, **16**(1), 2. 146 U. 1996. *The orientation*
*common vampire bats*. Royal 94 (<https://doi.org/10.1186/s128148>
Society Open Science, **4**(4), 95 98-016-0056-1) 147 *behaviour of the lesser*
160959. 96 25-26. Grant, G. S., & 148 *spearnosed bat, *Phyllostomus**
(<https://doi.org/10.1098/rsos>. 97 Bannack, S. A. 1999. *Harem* 149 *discolor (Chiroptera) in a*
160959) 98 *structure and reproductive* 150 *model roost: Concurrence of*
20-21. Dominoni, D.M., 99 *behaviour of *Pteropus** 151 *visual, echoacoustical and*
Halfwerk, W., Baird, E., 100 *tonganus in American* 152 *endogenous spatial*
Buxton, R.T., Fernández- 101 *Samoa*. Australian 153 *information*. Journal of
Juricic, E., Fristrup, K.M., 102 Mammalogy, **21**, 111–120. 154 Comparative Physiology A,
McKenna, M.F., Mennitt, 103 26-27. Gustin, M. K., & 155 **179**(2).
D.J., Perkin, E.K., Seymoure, 104 McCracken, G. F. 1987. 156 (<https://doi.org/10.1007/BF00>
B.M., Stoner, D.C., 105 *Scent recognition in the* 157 222791)
Tennessen, J.B., Toth, C.A., 106 *Mexican free-tailed bat,* 158 31-32. Kerth, G., Perony, N.,
Tyrrell, L.P., Wilson, A., 107 *Tadarida brasiliensis* 159 & Schweitzer, F. 2011. *Bats*
Francis, C.D., Carter, N.H., 108 *mexicana*. Animal Behaviour, 160 *are able to maintain long-*
& Barber, J.R. 2020. *Why* 109 **35**, 13–19. 161 *term social relationships*
*conservation biology can* 162 163 *despite the high fission-fusion*
164 *dynamics of their groups*.
Proceedings of Royal Society

- London B, **278**(1719), 2761–54
2767.
([https://doi.org/10.1098/rspb.](https://doi.org/10.1098/rspb.2010.2718)
2010.2718)
**32-33.** Kerth, G. 2008.
*Causes and consequences of*
*sociality in bats.* BioScience, **60**
**58**(8), 737–746.
(<https://doi.org/10.1641/B58062>
810)
**33-34.** Kilgour, R. J., Faure,
P. A., & Brigham, R. M.
2013. *Evidence of social*
*preferences in big brown bats*
*(Eptesicus fuscus).* Canadian
Journal of Zoology, **91**(10),
756–760.
([https://doi.org/10.1139/cjz-](https://doi.org/10.1139/cjz-2013-0057)
2013-0057)
**35.** Kohles, J.E., Carter,
G.G., Page, R.A., &
Dechmann, D.K.N. 2020.
*Socially foraging bats*
*discriminate between group*
*members based on search-*
*phase echolocation calls.*
*Behavioral Ecology, araa056,*
[https://doi.org/10.1093/beh-](https://doi.org/10.1093/beh-eco/araa056)
[eco/araa056](https://doi.org/10.1093/beh-eco/araa056)
**34-36.** Kunz, T. H., &
Fenton, M. B. 2003. *Bat*
*ecology.* University of
Chicago Press.
**35-37.** Lewis, S. E. 1995.
*Roost fidelity of bats: A*
*review.* Journal of
Mammalogy, **76**(2), 481–496.
([https://doi.org/10.2307/1382](https://doi.org/10.2307/1382357)
357)
**36-38.** Mann, O., Lieberman,
41 V., Köhler, A., Korine, C.,
Hedworth, H. E., & Voigt-
Heucke, S. L. 2011. *Finding*
*your friends at densely*
*populated roosting places:*
*Male Egyptian fruit bats*
*(Rousettus aegyptiacus)*
*distinguish between familiar*
*and unfamiliar conspecifics.*
Acta Chiropterologica, **13**(2)
411–417.
([https://doi.org/10.3161/1508](https://doi.org/10.3161/150811011X624893)
11011X624893)
37-39. Martens, M. J. M.
1980. *Foliage as a low-pass*
*filter: experiments with model*
*forests in an anechoic*
*chamber.* Journal of the
Acoustical Society of
America, **67**(1), 66–72.
**38-40.** Metheny, J. D.,
Kalcounis-Rueppell, M. C.,
Bondo, K. J., & Brigham, R.
64 M. 2008. *A genetic analysis*
*of group movement in an*
*isolated population of tree-*
*roosting bats.* Proceedings of
the Royal Society B:
Biological
Sciences, **275**(1648), 2265–
2272.
**39-41.** Mlinarić, A., Horvat,
73 M., & Šupak Smolčić, V.
2017. *Dealing with the*
*positive publication bias:*
*Why you should really*
*publish your negative results*
Biochemia Medica, **27**(3),
030201.
([https://doi.org/10.11613/BM](https://doi.org/10.11613/BM2017.030201)
2017.030201)
**40-42.** Møller, A. P., &
Jennions, M. D. 2001.
*Testing and adjusting for*
*publication bias.* Trends in
Ecology & Evolution, **16**(10),
580–586.
([https://doi.org/10.1016/S0169-](https://doi.org/10.1016/S0169-5347(01)02235-2)
5347(01)02235-2)
**41-43.** Moulton, D. G. 1967.
*Olfaction in mammals.*
American Zoologist, **7**(3),
421–429.
([https://doi.org/10.1093/icb/7.](https://doi.org/10.1093/icb/7.4.421)
4.421)
**42-44.** Mueller, H. C. 1966.
*Homing and distance-*
*orientation in bats.* PhD
thesis, University of
Wisconsin, Madison,
Wisconsin.
**43-45.** Murphy, M. 1993.
“Bats: A farmer’s best
friend.” Bats Magazine.
**11**(1).
**44-46.** Nelson, J. 1964.
*Vocal communication in*
*Australian flying foxes*
*(Pteropodidae;*
*Megachiroptera).* Zeitschrift
Fuer Tierpsychologie, **21**,
857–870.
**45-47.** Ober, HK. 2008.
*Effective bat houses for*
*Florida.* PhD thesis,
University of Florida Institute
of Food and Agricultural
Sciences, Gainesville,
Florida.
**46-48.** Partan, S. R., &
Marler, P. 2005. *Issues in the*
*classification of multimodal*
*communication signals.* The
American Naturalist, **166**(2),
231–245.
**47-49.** Popa-Lisseanu, A. G.,
Bontadina, F., Mora, O., &
IbÁñez, C. 2008. *Highly*
*structured fission–fusion*
*societies in an aerial-*
*hawking, carnivorous bat.*
Animal Behaviour, **75**(2),
471–482.
**48-50.** Ruczyński, I., &
Kalko, E. K. V. 2007. *The*
*sensory basis of roost finding*
*in a forest bat, Nyctalus*
*noctula.* Journal of
Experimental Biology,
**210**(20), 3607–3615.
([https://doi.org/10.1242/jeb.0](https://doi.org/10.1242/jeb.009837)
09837)
**49-51.** Ruczyński, I., Kalko,
E. K. V., & Siemers, B. M.
2009. *Calls in the forest: A*
*comparative approach to how*
*bats find tree cavities.*
Ethology, **115**(2), 167–177.
([https://doi.org/10.1111/j.143](https://doi.org/10.1111/j.1439-0310.2008.01599.x)
9-0310.2008.01599.x)
**50-52.** Safi, K., & Kerth, G.
2003. *Secretions of the*
*interaural gland contain*
*information about*
*individuality and colony*
*membership in the*
*Bechstein’s bat.* Animal
Behaviour, **65**(2), 363–369.
([https://doi.org/10.1006/anbe.](https://doi.org/10.1006/anbe.2003.2067)
2003.2067)
**51-53.** Schöner, C. R.,
Schöner, M. G., & Kerth, G.
2010. *Similar is not the same:*

- *Social calls of conspecifics* 40 *urban bushland*. *Biology* 78 60-62. Wilkinson, G. S.
*are more effective in* 41 *Letters*, **9**(3), 20121144– 79 1985. *The social organization*
*attracting wild bats to day* 42 20121144. 80 *of the common vampire bat:*
*roosts than those of other bat* 43 (<https://doi.org/10.1098/rsbl.2012.1144>) 81 *I. Pattern and cause of*
*species*. *Behavioral Ecology* 44 56-58. van Assen, M. A. L. 82 *association*. *Behavioral*
*and Sociobiology*, **64**(12), 45 83 *Ecology and Sociobiology*,
2053–2063. 46 M., van Aert, R. C. M., 84 **17**, 111–121.
(<https://doi.org/10.1007/s00247-010-1019-8>) 47 Nuijten, M. B., & Wicherts, 85 61-63. Wilkinson, G. S.,
52-54. Selvanayagam, P. F. 48 J. M. 2014. *Why publishing* 86 Carter, G. G., Bohn, K. M., &
10 L., & Marimuthu, G. 1984. 49 *everything is more effective* 87 Adams, D. M. 2016. *Non-kin*
*Spatial organization of* 50 *than selective publishing of* 88 *cooperation in bats*.
*roosting in the insectivorous* 51 *statistically significant* 89 *Philosophical Transactions of*
*tropical bat* *Hipposideros* 52 *results*. *PLOS ONE*, **9**(1), 90 *the Royal Society B:*
*speoris*. *Behavioural* 53 e84896. 91 *Biological Sciences*,
*Processes*, **9**(2), 113–121. 54 (<https://doi.org/10.1371/journal.pone.0084896>) 92 **371**(1687), 20150095.
53-55. Surlykke, A., Ghose, 55 57-59. Voigt, C. C., & von 93 (<https://doi.org/10.1098/rstb.2015.0095>)
17 K., & Moss, C. F. 2009. 56 Helversen, O. 1999. *Storage* 94 62-64. Willis, C. K., &
*Acoustic scanning of natural* 58 *and display of odor in male* 95 Brigham, R. M. 2004. *Roost*
*scenes by echolocation in the* 59 *Saccopteryx billineata* 96 *switching, roost sharing and*
*big brown bat*, *Eptesicus* 60 *Emballonuridae*. *Behavioral* 97 *social cohesion: forest-*
*fuscus*. *Journal of* 61 *Ecology and Sociobiology*, 98 *dwelling big brown bats,*
*Experimental Biology*, 62 **47**, 29–40. 100 *Eptesicus fuscus, conform to*
**212**(7), 1011–1020. 63 (<https://doi.org/10.1007/s00210-010-65005-6>) 101 *the fission–fusion model*.
(<https://doi.org/10.1242/jeb.024620>) 64 58-60. Wible, J. R., & 102 *Animal Behaviour*, **68**(3),
24620) 65 Bhatnagar, K. P. 1996. 103 495–505.
54-56. Theis, W., Kalko, E., 66 *Chiropteran vomeronasal* 104 63-65. Yohe, L. R., Davies,
& Schnitzler, H. 2016. *The* 67 *complex and the interfamilial* 105 K. T. J., Rossiter, S. J., &
*roles of echolocation and* 68 *relationships of bats*. *Journal* 106 Dávalos, L. M. 2019.
*olfaction in two neotropical* 69 *of Mammalian Evolution*, 107 *Expressed vomeronasal type-*
*fruit-eating bats*, *Carollia* 70 **3**(4), 285–314. 108 *I receptors (V1rs) in bats*
*perspicillata* and *C. castanea*, 71 <https://doi.org/10.1007/BF021077447> 109 *uncover conserved sequences*
*feeding on piper*. *Behavioral* 72 59-61. Wilkinson, G. S. 110 *underlying social chemical*
*Ecology and Sociobiology*, 73 1990. *Food sharing in* 111 *signaling*. *Genome Biology*
**42**(6), 397–409. 74 *vampire bats*. *Scientific* 112 *and Evolution*, **11**(10), 2741–
55-57. Threlfall, C., Law, B., 75 *American*, **262**(2), 76–82. 113 2749.
& Banks, P. B. 2013. *Odour* 76 *cues influence predation risk* 114 (<https://doi.org/10.1093/gbe/evz179>)
*at artificial bat roosts in* 77 115